# Effectiveness of Pain Neuroscience Education in Patients with Chronic Musculoskeletal Pain and Central Sensitization: A Systematic Review

**DOI:** 10.3390/ijerph20054098

**Published:** 2023-02-24

**Authors:** Beatrice Lepri, Daniele Romani, Lorenzo Storari, Valerio Barbari

**Affiliations:** 1Independent Researcher, 47921 Rimini, Italy; 2Department of Biomedical and Neuromotor Science, Bologna Campus, University of Bologna, 40138 Bologna, Italy; 3AUSL della Romagna, Ospedale Infermi di Rimini, Viale Luigi Settembrini, 2, 47923 Rimini, Italy; 4Department of Human Neurosciences, University of Roma “La Sapienza”, 00185 Rome, Italy

**Keywords:** pain neuroscience education, central sensitization, chronic musculoskeletal pain, systematic review

## Abstract

Objective: To collect the available evidence about the effectiveness of pain neuroscience education (PNE) on pain, disability, and psychosocial factors in patients with chronic musculoskeletal (MSK) pain and central sensitization (CS). Methods: A systematic review was conducted. Searches were performed on Pubmed, PEDro, and CINAHL, and only randomized controlled trials (RCTs) enrolling patients ≥18 years of age with chronic MSK pain due to CS were included. No meta-analysis was conducted, and qualitative analysis was realized. Results: 15 RCTs were included. Findings were divided for diagnostic criteria (fibromyalgia—FM, chronic fatigue syndrome—CFS, low back pain—LBP, chronic spinal pain—CSP). PNE has been proposed as a single intervention or associated with other approaches, and different measures were used for the main outcomes considered. Conclusions, practice implication: PNE is effective in improving pain, disability, and psychosocial factors in patients with fibromyalgia, chronic low back pain (CLBP)—especially if associated with other therapeutic approaches—and also in patients with CFS and CSP. Overall, PNE seems to be more effective when proposed in one-to-one oral sessions and associated with reinforcement elements. However, specific eligibility criteria for chronic MSK pain due to CS are still lacking in most RCTs; therefore, for future research, it is mandatory to specify such criteria in primary studies.

## 1. Introduction

### 1.1. Background

Chronic musculoskeletal (MSK) disorders are one of the main health problems worldwide [1], with chronic low back pain (CLBP) still being the most common [2,3,4]. In detail, chronic pain can be defined as pain lasting beyond the normal healing time [5], beyond 3 months [6], or, lastly, beyond 3–6 months [7], according to different definitions and criteria arising from previous papers. Chronic MSK pain should also be divided into primary or secondary chronic MSK pain. Primary chronic MSK pain—the object of interest in this review—is considered the *“chronic pain experienced in muscles, bones, joints, or tendons that (1) is characterized by significant emotional distress (such as anxiety, anger, frustration, or depressed mood) or functional disability (interference in daily life activities and reduced participation in social roles), and (2) cannot be attributed directly to a known disease or damage process”* [8]. Despite the variability of chronic MSK disorders between countries, the estimated prevalence is still high ranging from 11.4% to 24% [9].

Interestingly, most of the patients suffering from MSK pain are unaware of the proper pathway of care to undertake. Therefore, there is a negative impact on pain, disability, and quality of life, with a direct increase in healthcare-related costs for these patients [10,11]. For these reasons, the importance of a biopsychosocial approach has progressively been highlighted for the management of persistent MSK pain, and the most recent approaches include therapeutic exercise, manual therapy associated with exercise, pharmacological pain management, and patient education [12,13,14,15,16,17].

Among the most widespread educational techniques, pain neuroscience education (PNE) has been recognized as an effective approach for the management of patients with persistent MSK disorders, showing clinically relevant results on pain, disability, and psychosocial factors—especially as an adjunct to exercise and/or manual therapy [18,19,20,21,22]. Several authors established that PNE is aimed at reconceptualizing pain perceptions, beliefs, and illness or avoidance behaviors through educational sessions such as one-to-one or collective oral sessions, phone calls, or written materials (booklets or email) [23,24]. However, PNE is not applicable to all patients, and some limits in clinical practice have already been previously highlighted [25]. In 2011, John Nijs and colleagues developed guidelines for the administration of PNE in MSK practice [24]. Specifically, they recommend PNE approaches in two cases: (1) patients with a medical diagnosis consistently related to a dominance of central sensitization (CS) or (2) patients with maladaptive coping strategies, illness perceptions or behaviors, and pain beliefs. Furthermore, to improve the specificity in the identification of patients who may benefit from PNE, the same authors have already provided detailed guidelines for the identification of those with a dominant CS pain mechanism [24,26].

Although PNE seems to be an effective intervention for patients with persistent MSK pain [19], data about the effectiveness of PNE in specific populations of patients with persistent MSK pain due to CS are sparse. In such a scenario, most of the inclusion criteria of published primary or secondary studies are limited to the general chronic MSK pain, but no specific eligibility criteria for CS have been addressed. Given that CS has clearly defined features [26], the effectiveness of PNE in patients with chronic MSK pain due to a dominant CS mechanism still remains a grey area of scientific literature [20,27,28].

### 1.2. Objectives

The primary aim of this systematic review is to collect the available evidence concerning the effectiveness of PNE, specifically in individuals suffering from chronic MSK pain and CS, on clinically relevant outcomes and to provide recommendations for clinicians and upcoming research.

## 2. Methods

This systematic review was conducted based on the PRISMA Statement 2020 [29]. The protocol was registered with PROSPERO [30] (CRD42022356005).

### 2.1. Eligibility Criteria

#### 2.1.1. Study Design

Only randomized controlled trials (RCTs) published in Italian or in English were considered eligible. No further restrictions were applied.

#### 2.1.2. Participants

Studies enrolling patients ≥18 years of age with persistent MSK pain for at least 3 months due to CS were included [6]. In detail, to increase the specificity of CS of the included patients, RCTs were included if their participants were in line with diagnostic criteria related to CS pain mechanism in the literature [24,26,31,32]. For this reason, the study selection used: central sensitization inventory (CSI) scores > 40 and quantitative sensory testing (QST) positive scores for CS or any other criteria [such as fibromyalgia or chronic fatigue syndrome (CFS), or psychosocial factors, maladaptive pain beliefs, illness behaviors]. Furthermore, if studies did not specify such aspects in their eligibility criteria (e.g., only patients with chronic MSK pain without any other information), an in-depth analysis was performed of the baseline characteristics of participants to identify any information that may be related to CS dominance. To make such screening, all questionnaires (such as the CSI itself or Pain Catastrophizing Scale—PCS, Fear-Avoidance Beliefs Questionnaire—FABQ, Tampa Scale of Kinesiophobia—TSK or other measures and questionnaires) were examined through a manual screening. Patients with scores in accordance with a dominance of CS pain mechanism diagnosis were included in the present study.

RCTs were excluded if they enrolled patients aged <18 years of age; or acute, subacute, or recurrent pain conditions or with pain of any duration caused by specific pathologies (pulmonary, cardiac, neurological, oncological, visceral, cognitive, psychiatric disorders) or patients who had surgical back procedures within a year.

No restrictions in terms of publication date have been implemented.

#### 2.1.3. Interventions

RCTs were eligible if interventions were based on PNE proposed in any format. No restrictions were applied in terms of the combination of PNE with other interventions.

#### 2.1.4. Comparisons

Educational interventions, waiting lists, placebo interventions, or other active (e.g., exercise) or passive (e.g., manual therapy) approaches were eligible for inclusion.

#### 2.1.5. Outcome and Outcome Measures

To be included, RCTs had to assess at least 1 of the following outcomes: (1) pain, (2) disability, and (3) psychosocial factors. No restrictions were applied in terms of outcome measures.

### 2.2. Search Methods for Inclusion of Studies

#### Electronic Searches

An electronic search was conducted between May and September 2021 on the following databases: PubMed, PEDro, and CINAHL. Searches were set and managed according to the specific settings of each database. Search strings were composed using MESH (Medical Subject Headings)—where possible—or free terms and combined with Boolean operators (AND, OR, and NOT) in line with the PI(C)O model of clinical questions (participants, interventions, outcomes). An additional search on the main 3 databases was conducted between September and October 2022 to add also papers published after the first round of search. Furthermore, all bibliographies of the included studies and all other existing systematic reviews focused on PNE were manually screened to identify other potentially relevant papers.

The full search strategy for PubMed is available in Appendix A.

### 2.3. Study Selection and Data Extraction

After the removal of duplicates, all records were screened for title and abstract. Then, all full-text articles suitable for inclusion were screened according to the inclusion criteria of the present study through the independent analysis of the main two authors (B.L. and V.B.). A third author (L.S.) not involved in the screening process was involved in case of disagreements. The data extraction process was performed independently by the two main authors (B.L. and V.B.) Full-text papers were retrieved thanks to the Library Service of the University of Bologna and through direct e-mail contact with authors.

### 2.4. Inter-Rater Agreement

To quantify the inter-rater agreement between the two main authors for full-text selection, Cohen’s Kappa (K) was used. K value was calculated and interpreted according to Altman’s definition [33]: low (k < 0.2), fair (0.2 < k < 0.4), modest (0.41 < k < 0.61), good (0.61 < k < k0.80), excellent (x > 0.80).

### 2.5. Risk of Bias

The risk of bias in the included studies was independently assessed by the two main authors (B.L. and V.B.) through the Risk of Bias (RoB) assessment tool of the Cochrane Collaboration [34], and a third author (L.S.) was involved in case of disagreements.

### 2.6. Analysis

Due to the high heterogeneity across the included studies, no meta-analysis was performed, and a qualitative synthesis was conducted in a narrative and tabular format. All data from the included studies related to both between-groups and within-group differences were reported for each outcome. Where possible, punctual estimates, confidence intervals, standard deviation, effect size, statistical significance (*p*), and clinical relevance (minimal clinical important difference –MCID– or any other measure) were reported.

## 3. Results

In total, 262 records were retrieved throughout the electronic searches. After the removal of duplicates, title, and abstracts screening led to 143 potentially relevant articles. Finally, a further 128 articles were excluded after the full-text screening, and 15 articles satisfied the inclusion criteria and were included in this systematic review.

The full selection process is reported in Figure 1.

### 3.1. Study Characteristics

Overall, 15 RCTs were included [35,36,37,38,39,40,41,42,43,44,45,46,47,48,49]. All characteristics (study design, recruitment, age, sex, duration of pain, diagnostic criteria, and the number of participants) are described in detail in Table 1. Particularly, in 13 studies, participants were divided into 2 intervention groups [35,36,37,38,40,41,42,43,44,46,47,48,49], while 2 studies were multiple-arms RCTs [39,45]. Furthermore, three studies were multicenter trials [35,39,48].

#### 3.1.1. Sample

In total, the participants included and randomized were 1085 (220 males, 863 females, and 2 others). Across all RCTs, the study of Téllez et al. [47] in 2015 had the smallest sample size (12), and the study of Serrat et al. [46] in 2020 had the largest sample size (169).

#### 3.1.2. Drop-Out and Lost to Follow-Up

Out of 1085 patients recruited, there were 60 (5.53%) drop-outs, and 77 (7.10%) lost to follow-up. Details are specified in Table 2.

#### 3.1.3. Follow-Ups

The timing of re-assessment significantly varied across the included studies. Follow-ups ranged between a minimum of after-treatment follow-up [40] and a maximum of 12 weeks follow-up [35]. Details are reported in Table 1.

#### 3.1.4. Adverse Effects

Only two studies [42,43] have specified the absence of adverse events, and one study [35] did not specify adverse events.

#### 3.1.5. Type of Participants

The mean age of all participants was 45.92 years. Although all participants suffered from persistent MSK pain, diagnostic labels significantly differed. Six studies [37,41,42,43,44,47] included patients with CLBP lasting for more than 3 months; two studies [36,45] enrolled patients with CLBP lasting for more than 6 months; one study [38] included patients with fibromyalgia (FM) and CLBP; one study [39] enrolled patients with CSP; one study [40] included patients diagnosed with chronic CFS defined by the Centers for Disease Control and Prevention criteria [51], and four studies [35,46,48,49] included patients with fibromyalgia based on the American College of Rheumatology (ACR) criteria [50]. All specific characteristics are listed in Table 1.

#### 3.1.6. Type of Interventions

PNE was proposed in different modalities in all experimental interventions both alone and in association with other therapeutic approaches. In particular, PNE was proposed as a single intervention [38,39,40,44,48,49], “sensitized” (culture-sensitive PNE approach, based on pain-related beliefs, cognitions, and behaviors of Turkish patients, adapted from rounds of a previous Delphi study) [41] or associated with other therapeutic approaches such as physiotherapy [37], therapeutic exercise [36,43], water-based exercises [42], manual therapy and home exercises [45], dry needling [47], usual care [35], other types of education with therapeutic exercise and outdoor activities associated with usual treatment [46]. Moreover, PNE administration modalities also differed in terms of duration and frequency of the treatment, assigned staff, topics treated, and instruments used. Details are specified in Table 1.

#### 3.1.7. Type of Control Groups

Participants in the control group were subjected to different approaches such as self-management education [40,49], education and relaxation [48], neck/back school [39], usual care [35,46], health behavior control [38], dry needling [47], physiotherapy [37], therapeutic exercise [36], water exercises [42], group exercises [43], home exercises alone or in combination with manual therapy [45]. PNE is also administered in the control group in two studies: a standard approach [41] and PNE + aerobic exercises [44]. Specificities for each control group are expressed in Table 1.

#### 3.1.8. Type of Outcome and Outcome Measures

Significant heterogeneity of outcome measures was found across the included studies for pain, disability, and psychosocial factors. Specifically, pain was assessed with NPRS in three studies [36,45,47], NRS in two studies [41,44], and VAS in three studies [38,43,46]. Disability was evaluated with RMDQ in six studies [36,37,41,43,44,47], QBPDS-PT in one study [42], ODI in two studies [45,47], FIQ in three studies [35,48,49], FIQR in one study [46]. Psychosocial factors were assessed with QBPDS-PT in 1 study [42], PCS in 9 studies [35,36,38,39,40,41,46,48,49], TSK in 12 studies [36,37,38,39,40,41,42,44,45,46,47,49], BPI in 2 studies [35,38], PCI in 2 studies [40,49], PBQ in 1 study [41], PVAQ in 2 studies [39,49], HADS in 2 studies [35,46], HAQ in 1 study [35], PDI in 1 study [39], in 2 studies IPQr [39] and IPQ-FM [48], RSES in 1 study [46], PROMIS in 1 study [38], FABQ in 1 study [43], PSEQ in 2 studies [43,44], PSCOQ in 1 study [38], SF-36 in 2 studies [46,49], and SWLS in 1 study [38]. All details are listed in Table 3.

### 3.2. Risk of Bias

Selection bias was low across the included studies except for two RCTs [37,41] with unclear information about the randomization procedure. The performance bias was rated as high risk in all 15 studies included. Since seven studies [39,40,42,43,45,47,48] provided effective measures to ensure the blindness of assessors, a low risk for attrition bias was attributed. In contrast, in five articles [35,36,37,44,49], the blindness of the evaluators was not guaranteed, leading to a high risk of bias. Three studies [38,41,46] showed an unclear risk for detection bias. All studies were evaluated with a low bias risk for the incomplete reporting of data, except for three studies [37,39,46] that showed a lack of details. For reporting bias, only two studies [35,36] were at high risk, one study [37] was at unclear risk, and all the other studies were judged at low risk. In the evaluation of other biases, two studies [35,46] were labeled with low risk, three studies [44,46,47] with a high risk, and the remaining studies were at unclear risk. All details are listed in Table 4.

### 3.3. Agreement

The inter-rater agreement index (B.L. and V.B.) was good (K = 0.70) for full-text selection. Data are detailed in Table 5.

### 3.4. Effects of Interventions

The qualitative synthesis for the effectiveness of PNE was divided for the diagnostic label (fibromyalgia, CSP, CFS, fibromyalgia and/or CLBP and CLBP alone) and reported both in a narrative and a tabular format. Details are reported in Table 6.

#### 3.4.1. Fibromyalgia

In the treatment of patients with fibromyalgia, PNE had similar clinical results in improving all the considered outcomes if compared with education and relaxation techniques [48]. Similar conclusions were drawn for PNE when compared with education and self-management, except for the SF-36 questionnaire, where a significant improvement was observed in the intervention group with PNE [49]. When compared to usual treatment [35], PNE was more effective in improving outcome measures such as FIQ, BPI, HAQ, HADS, and PCS. When proposed in a multimodal program (TAU + NAT − FM), PNE was superior to usual care (TAU) for all outcome measures except for RSES [46]. Details are reported in Table 6.

#### 3.4.2. Chronic Spinal Pain (CSP)

In the single RCT focused on CSP, PNE was more effective than education based on neck/back school in improving PCS values in participants with high CSI scores and improving PDI. In contrast, TSK-17 and IPQR values in both intervention groups, regardless of CSI level, were observed [39]. Details are reported in Table 6.

#### 3.4.3. CFS

Compared with the self-management education of ADL [40], PNE was more effective in improving PCS “rumination,” PCI “distraction,” and PCI “worrying.” There were no significant differences between groups in the other PCI and PCS domains and in TSK scores. Details are reported in Table 6.

#### 3.4.4. Fibromyalgia and/or CLBP

At the 1-month follow-up, PNE was significantly superior to the “Health Behavior Control” program on PSCOQ and BPI (severity and interference) scores [38]. However, the subsequent assessment at the 10-month follow-up showed no difference in all outcome measures. Details are reported in Table 6.

#### 3.4.5. CLBP

In the study of Bodes et al. [36] in 2018, PNE associated with therapeutic exercise was more effective than therapeutic exercise alone in improving TSK-11 and PCS-13 at both follow-ups, NPRS and RMDQ at 3 months. Conversely, in the study by Pires et al. [42] in 2014, PNE associated with aquatic exercises showed similar results as aquatic exercises alone. There were no significant differences between the intervention group and the control group either between physiotherapy associated with PNE against physiotherapy alone [37] or between PNE sensitized against the standard PNE, although in the latter case, significant improvements over time in both groups with the two types of PNE were registered [41]. For VAS and RMDQ scores at the 8-week follow-up, a significant group difference was found in favor of the intervention group (PNE only) versus PNE associated with aerobic exercise [43]. At the same follow-up, PNE only associated with aerobic exercises was more effective in improving pain (NPRS) and psychosocial factors (PSEQ), losing significance values at the 3-month follow-up [44]. PNE combined with dry needling (DN) also was more effective than DN alone in improving TSK-17 scores, reaching the clinically significant difference (MCID > 8) [47]. In the study by Saracoglu et al. [45] in 2020, participants were divided into three groups, each with 23 participants. There was a significant improvement in TSK-17 scores at both follow-ups in the first group (PNE, manual therapy, and exercises at home) against the second group (manual therapy and exercises at home). In the first group compared with the third one (home exercises), there were significant improvements in TSK-17, NPRS, and ODI values at both follow-ups. Details are reported in Table 6.

## 4. Discussion

The objective of this SR was to collect the available evidence about the effectiveness of PNE in chronic MSK patients due to CS on pain, disability, and psychosocial factors.

The risk of bias in most of the studies was low, except the performance bias criterion rated as high risk in all studies. Nevertheless, since a low risk for performance bias may be hard to obtain in physical therapy trials, it does not seem to be a significant factor in downgrading the overall quality of evidence.

Overall, PNE seems to be effective both as a single intervention and more effective if proposed in a multidisciplinary program. Furthermore, only in two studies [37,48] was PNE not significantly superior to controls on all outcome measures. However, since most of the follow-ups were established in the short- or medium-term with few studies addressing long-term follow-ups, it is still unclear if such promising results supporting the effectiveness of PNE (or the combination of PNE with other effective therapeutic approaches) in chronic MSK patients with a dominant CS pain mechanism may also be maintained in the long-term.

### 4.1. Most Effective Strategies for Diagnostic Label

#### 4.1.1. Fibromyalgia

In the context of FM, PNE proposed in combination with usual treatments for FM was more effective than usual treatments only, as appreciated in two studies with a low to moderate risk of bias [35,46]. However, in one RCT [35], the authors only provided *p*-values for FIQ scores (*p* < 0.001), and no data were specified for the remaining so-defined (in the results section) “significant” differences. At all follow-ups (1, 6, and 12 months), there were also improvements defined by authors as “clinically relevant,” but no data for clinical relevance were provided. The second study [46] supports such results revealing the superiority of PNE associated with usual treatments in all outcome measures except for RSES scores. Findings rising from the other 2 moderate to high-quality RCTs [48,49] do not support the effectiveness of PNE as a stand-alone intervention—if compared to education and relaxation [48] or self-management instruction [49]—both in the short- and medium-terms. The discrepancy in results of PNE in FM patients may be related to the necessity of such treatment to be involved in a more comprehensive approach for chronic MSK pain.

#### 4.1.2. CSP

For CSP patients [39], results with a low risk of bias supporting the effectiveness of PNE over neck/back school education are limited to short-term follow-ups (2 weeks), and no long-term benefits were assessed. Although CSI scores were not assessed at the end of the study, it is noteworthy that there has been an improvement in outcomes closely related to CS, such as kinesiophobia and perception of disease, regardless of the level of CSI used to divide the participants in the baseline.

#### 4.1.3. CFS

For CFS patients, conflicting results arising from the unique moderate-quality RCT [40] are strictly limited to the short-term (immediately post-session), and only psychosocial factors were evaluated. For the same outcome measures (e.g., PCS), different results were obtained (e.g., rumination and magnification or helplessness). Such contradictory results are far more reasonable since a single PNE session may not be effective enough (or more effective than other educational approaches) to modify beliefs, perceptions, and thoughts. The latter consideration is in line with Nijs and colleagues, who recommend at least two educational sessions: the first intended to explain CS and pain neurophysiology, and the second aimed at making sure patients understand previous pain [24,26].

#### 4.1.4. Fibromyalgia and/or CLBP

When the participants are both FM and CLBP patients [38], PNE is an effective strategy to improve all psychosocial factors. Indeed, the only results supporting the effectiveness of PNE over a health behavior control approach are limited to the short-term (1 month). Such findings are not surprising since it is unlikely that a single PNE session lasting for 20–25 min with a 3-min instructional video (experimental intervention) will be more effective than the same procedure (control intervention) without significant differences in terms of educational contents.

#### 4.1.5. CLBP

Overall, among the studies included with patients suffering from CLBP, findings from the included RCTs support the effectiveness of PNE, especially if associated with other therapeutic approaches. However, since results differed in terms of follow-up and outcome measures, further considerations are needed. Firstly, it seems that the PNE procedure does not affect outcomes, and both “culture-sensitive” and “standard” PNE are effective in the short term [41]. Significantly better short-term results were obtained if PNE was associated with exercise therapy [36] but not with usual physiotherapy [37]. This is in line with previous literature supporting the effectiveness of different forms of exercise in the management of CLBP patients [52,53,54,55]. However, the type of exercise therapy seems to be crucial. The combination of PNE and aquatic exercise did not add better results than aquatic exercise alone—except for VAS scores at 3 months [42]. The combination of PNE and dry needling (DN) compared to DN was more effective only on TSK scores but not on disability and pain measures. The latter aspect may be due to the reduction of fear of movement thanks to PNE concepts, which do not directly address pain or disability. Furthermore, such results must be interpreted with caution. Since DN is not a recommended procedure for CLBP patients, it does not represent a major comparator for the investigation of PNE effectiveness in CLBP. Surprisingly, previous findings are not in line with those rising from the comparison of PNE versus PNE + aerobic circuit-based exercise [44] and PNE + manual therapy (MT) and home-based exercise (HEX) versus two groups (MT + HEX and HEX only) [45]. In such a scenario, it is still unclear if the latter results are mainly related to the PNE concepts, administration modalities, type of MT approach, or exercise modalities. 

### 4.2. Applicability of Results and Training for Health Professionals

RCTs included in this SR support the effectiveness of PNE in patients with chronic MSK pain due to CS, but the applicability of these results is questionable.

In terms of PNE administration, several modalities were proposed across the included RCTs. Furthermore, PNE was more effective throughout one-to-one oral sessions rather than the group- or online- or written-based approaches, and such findings are surely a major strength of PNE and its applicability in clinical practice. In terms of health professionals, PNE administration needs specific training to achieve enough knowledge of contents regarding neuroanatomy, neurophysiology, and pain mechanisms. Explanations proposed to patients must be proposed using understandable and simple language, metaphors, and examples—requiring, again, specific communicative- and educative-based training. Since not all clinicians are confident and trained in communicative skills and pain education, the latter aspect may limit the applicability of promising results of PNE in clinical practice. Moreover, PNE needs to be proposed only to selected patients; therefore, all clinicians should be able to screen patients in terms of pain mechanism and to recognize all the psychosocial factors related to CS—limiting the transferability of results of this systematic review only to those clinicians trained in these terms. Finally, nothing is known about the long-term effectiveness of PNE. The majority of follow-ups were set in the short- and medium-term. For this reason, the applicability of results in the long term is not possible.

Based on our findings, an urgent need for specific training in the context of pain education is needed, which should be gathered with other mandatory skills for clinicians in direct-access practice, such as screening for referral [56,57,58,59], knowledge in pain neuroscience [25], and other domains of rehabilitation such as exercise [60,61,62].

### 4.3. Consistency

Since our SR specifically addressed the effectiveness of PNE in patients with chronic MSK pain and CS, the consistency of our findings was limited to previous papers that investigated the overall effectiveness of PNE on MSK pain.

Overall, the results of this SR are consistent with other previous papers. Indeed, PNE already showed effective results in patients with chronic MSK pain, especially when included in multimodal programs [19]. Previous authors demonstrated that the combination of PNE and exercise, compared to exercise alone, leads to greater short-term improvements in pain, disability, and psychosocial factors [63]. Similar conclusions were drawn by Wood and colleagues, supporting the effectiveness of PNE on disability in the short term if combined with the usual-care physiotherapy treatments for CLBP [20]. Finally, Bülow and colleagues, in the first meta-analysis focused on the effectiveness of PNE, showed the promising effects of PNE on pain, disability, and physical, psychological, and social function [64], despite the low quality of included studies.

Therefore, all results gathered from previous papers are consistently in line with our findings. However, none of the previous SRs specifically focused on chronic MSK pain and CS; for this reason, consistency is still limited, and further research is needed.

### 4.4. Strengths and Limitations

The first limitation of this SR is the lack of specificity of inclusion criteria of primary studies related to the sample of participants with a dominant CS pain mechanism. Indeed, only two studies [36,39] clearly declared a sample of patients with chronic pain and CS using CSI scores. Although the main authors of this systematic review (B.L. and V.B.) screened all baselines characteristics of participants in all RCTs to improve the specificity of the participants with a dominant CS pain mechanism, the latter aspect must be intended as a significant limitation for both this work and current research in the field of PNE and CS. However, although studies enrolling patients with FM [35,38,46,48,49] or CFS [40] did not use specific inclusion criteria for CS, such conditions are currently considered pathologies with a dominant CS pain mechanism [26]. Therefore, such studies might extend the research focused on PNE and CS despite the lack of specific inclusion criteria.

To the best of the authors’ knowledge, this is the first systematic review focused on the effectiveness of PNE in patients with chronic MSK pain and CS; this is a major strength of this work. Furthermore, the accurate research screening process on databases, the manual searches, the double-blind assessment for the risk of bias across the studies, and the good Cohen’s K score for full-text selection are strengths of the methodological quality of this SR.

## 5. Conclusions

Overall, PNE effectively improves pain, disability, and psychosocial factors in patients with persistent MSK pain and CS. In particular, the one-to-one modality with medium- or long-term sessions with reinforcements (brochures and comprehension exercises) seems to be more effective than remote sessions (telephone and computer) or content-only reading.

In patients with FM, PNE showed promising results when included in a multidisciplinary program, compared with usual care, but not in comparison with other educational or self-management techniques. In patients with CSP and CFS, PNE seems reliable to improve clinical outcomes in the short term when proposed individually against other educational or self-management approaches. Moreover, in patients with CLBP, PNE appears to be effective in combination with other treatments, such as manual therapy and mostly therapeutic exercise.

### 5.1. Implications for Practice

PNE is a valuable and effective intervention for clinicians in the treatment of patients with persistent MSK pain and a dominant CS pain mechanism. Furthermore, the effectiveness of PNE is more evident if proposed in association with other therapeutic modalities such as manual therapy, therapeutic exercise, and self-management strategies.

### 5.2. Implications for Research

It is mandatory for future research to homogenize the inclusion criteria in RCTs in order to improve the specificity of sub-groups of participants with a dominance of the CS pain mechanism. Furthermore, longer follow-ups are needed to assess the long-term benefits of PNE, and more standardization of PNE procedures proposed for chronic MSK with CS is needed to summarize these results in a future meta-analysis quantitatively.

## Figures and Tables

**Figure 1 ijerph-20-04098-f001:**
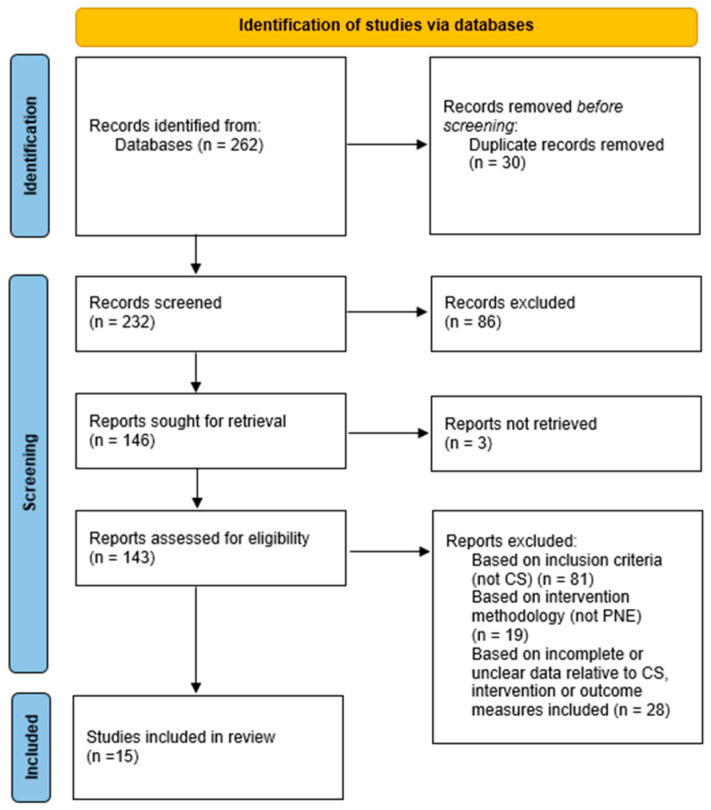
Flow chart and inclusion process of primary studies.

**Table 1 ijerph-20-04098-t001:** Characteristics of the included studies.

General Informations (Author, Years, Study Design, Country)	Partecipants (Characteristics, Number, Age, Gender, Recriument)	Interventions (Number of Participants, Content, Frequency, Duration and Professional in Charge)	Comparisons (Number of Participants, Content, Frequency, Duration and Professional in Charge)	Outcome Measures and Follow-Up	Results (EX: Experimental Group; C: Control Group)
Barrenengoa-Cuadra et al. [35], in 2021 RCT Spain	Fibromyalgia according to American College of Reumatology criteria (ACR) [50] N = 140 Age: > 18 years Patients recruited from waiting lists at five medical centres in Bilbao, using an electronic database.	Number (N) = 70, Male (M) = 2, Female (F) = 68, Age = 52,3 ± 9,2 1- Pain Neuroscience Education (PNE): 6 weekly lessons lasting 2 h, plus a 7° session as reinforcement after 1 month. The intervention involved the use of audio-visual material to treat concepts related to the neurobiology of pain and movement awareness. The material was also sent by email to patients after each session. Professional in charge: multidisciplinary team composed of 2–3 physiotherapists specialized in educational interventions on fibromyalgic patients 2- Usual care (same as control group)	N = 69, M = 6, F = 63, Age = 51.4 ± 10.2 1- Usual care: In Spain, the usual treatments include use of drugs, the most used are antidepressants, antiepileptics and analgesics. Professional in charge: doctors for prescription of medicines.	*Fibromyalgia Impact Questionnaire* (FIQ) *Brief Pain Inventory* (BPI-SF) *Health Assessment Questionnaire* (HAQ) *Hospital Anxiety and Depression Scale* (HADS) *Pain Catastrophizing Scale* (PCS-13) Baseline 1 month 6 months 12 months	FIQ EX: 60.9 ± 15.3 ⟶ 36.5 ± 21.8 ⟶ 38.0 ± 24.2 ⟶ 37.4 ± 24.1 C: 60.1 ± 13.8 ⟶ 60.6 ± 12.8 ⟶ 57.0 ± 14.3 ⟶ 56.7 ± 15.6 *p* < 0.001 BPI-SF (severity) EX: 5.8 ± 1.7 ⟶ 3.5 ± 1.9 ⟶ 4.0 ± 2.2 ⟶ 3.7 ± 2.2 C: 5.6 ± 1.7 ⟶ 5.7 ± 1.7 ⟶ 5.2 ± 1.9 ⟶ 5.5 ± 1.8 *p* < 0.05 BPI-SF (interference) EX: 6.6 ± 2.2 ⟶ 3.5 ± 2.7 ⟶ 3.6 ± 2.7 ⟶ 3.5 ± 2.7 C: 6.4 ± 2.2 ⟶ 6.6 ± 2.2 ⟶ 5.7 ± 2.2 ⟶ 5.9 ± 2.3 *p* < 0.05 HAQ EX: 1.4 ± 0.5 ⟶ 0.6 ± 0.5 ⟶ 0.6 ± 0.5 ⟶ 0.7 ± 0.5 C: 1.3 ± 0.6 ⟶ 1.1 ± 0.6 ⟶ 1.0 ± 0.6 ⟶ 1.1 ± 0.6 *p* < 0.05 HADS (anxiety) EX: 13.1 ± 3.9 ⟶ 8.4 ± 4.6 ⟶ 8.7 ± 4.9 ⟶ 8.2 ± 4.2 C: 12.3 ± 4.2 ⟶ 11.8 ± 4.3 ⟶ 11.8 ± 4.1 ⟶ 11.9 ± 4.1 *p* < 0.05 HADS (depression) EX: 9.5 ± 4.4 ⟶ 1.3 ± 0.9 ⟶ 4.9 ± 4.5 ⟶ 5.1 ± 4.9 C: 9.2 ± 4.0 ⟶ 2.4 ± 2.4 ⟶ 9.3 ± 4.5 ⟶ 8.8 ± 4.7 *p* < 0.05 PCS-13 EX: 26.9 ± 14.6 ⟶ 11.0 ± 11.3 ⟶ 11.6 ± 12.8 ⟶ 10.6 ± 12.3 C: 24.4 ± 13.0 ⟶ 25.5 ± 15.7 ⟶ 23.8 ± 14.2 ⟶ 23.3 ± 15.5 *p* < 0.05
Bodes et al. [36], in 2018 RCT Spain	CLBP > 6 months (patients with high CSI values at baseline) N = 56 Age: 20–75 Patients recruited through advertisements posted in four clinics and in the University of Alcalà in Madrid	N = 28, M = 6, F = 22, Age = 44.9 ± 9.6 1- PNE: 2 educational sessions of 30–50 min applied on patients divided into groups of 4–6 people. First session: explanation of the concepts underlying the neurophysiology of pain and delivery of a booklet to be read to reinforce the information shared. Second session (after 1 month): in-depth discussion and discussion of the contents of the first session. Professional in charge: physiotherapist expert on PNE. 2- Therapeutic exercise (same as control group)	N = 28, M = 6, F = 22, Age = 49.2 ± 10.5 1- Therapeutic exercise: proposed motor control exercises, stretching and aerobic exercises First session: Explanation of exercises to patients and supervision during execution in order to make them autonomous in following the plan at home. Exercises are done every day for 3 months Second session (after 1 month): control and correction of the exercises assigned in the first session. Professional in charge: physiotherapist experienced in motor control.	*Numerical Pain Rating Scale* (NPRS) *Roland Morris Disability* Questionnaire (RMDQ) *Pain Catastrophizing Scale* (PCS-13) *Tampa Scale of Kinesiophobia* (TSK-11) Baseline 1 month 3 months	NPRS EX: 7.9 [7.4,8.4] ⟶ 3.9 [3.2,4.6] ⟶ 2.7 [2.0,3.4] C: 7.8 [7.5,8.4] ⟶ 6.0 [5.4,6.6] ⟶ 4.8 [4.1,5.5] *p* < 0.001 3 months RMDQ EX: 12.0 [11.4,12.6] ⟶ 8.5 [7.8,9.3] ⟶ 6.4 [5.5,7.2] C: 12.6 [12.1,13.1] ⟶ 11.0 [10.3,11.6] ⟶ 9.8 [8.9,10.6] *p* < 0.001 3 months PCS-13 EX: 34.1 [31.2,37.0] ⟶ 22.2 [18.8,25.6] ⟶ 18.2 [15.4,21.0] C: 32.1 [30.2,34.1] ⟶ 28.7 [26.6,30.8] ⟶ 26.9 [24.8,29.0] *p* < 0.001 3 months TSK-11 EX: 28.7 [26.1,30.9] ⟶ 20.1 [18.5,21.6] ⟶ 16.1 [15.2,16.9] C: 28.1 [26.0,30.2] ⟶ 26.1 [24.1,28.0] ⟶ 24.1 [22.0,26.1] *p* < 0.001 3 months
Gül et al. [37], in 2021 RCT Turkey	CLBP > 3 months N = 31 M = 5, F = 26 Age: 18–60 Patients recruited from the clinic in Antalya	N = 16, Age: 42.1 ± 10.1 Both groups followed 15 sessions of physiotherapy, 3 each week. 1-TNE: 2 sessions per week, lasting 40 min. Conducting a one-to-one interview in a quiet and illuminated room; the arguments explained in the first session were related to neurophysiological mechanisms and psychosocial factors underlying pain. Images, metaphors, drawings were used and participants were given a brochure to read at home and the physiotherapist asked questions about the contents of the first session as reinforcement. Professional in charge: physiotherapist specialized on the TNE concept, as described by Moseley and Butler. 2- Physiotherapy (same as control group)	N = 15, Age: 42.5 ± 12.0 1- Physiotherapy: includes the use of: Hot-pack (20 min), ultrasound (10 min, intensity of 1,5 watt/cm^2^, frequency of 1 Mhz), TENS (at the beginning 80 μsec/100 Hz), successive sessions with acupuncture TENS (200 μsec/5 Hz for 20 min). Physiotherapist delivered a written program of exercises after teaching them to all patients. The exercise plan includes isotonic and isometric reinforcement, exercises for trunk muscles, stretching. Compliance was monitored with a diary. Professional in charge: physiotherapist	*Visual Analogue Scale* (VAS) *Tampa Scale of Kinesiophobia* (TSK-11) *Roland Morris Disability* Questionnaire (RMDQ) Baseline 3 weeks	VAS EX: Δ ⟶ −35.9 ± 28.3 C: Δ ⟶ 33.8 ± 29.5 *p* > 0.05 TSK-11 EX: Δ ⟶ −17.3 ± 12.1 C: Δ ⟶ −2.9 ± 6.4 *p* = 0.410 RMDQ EX: Δ ⟶ −8.8 ± 5.5 C: Δ ⟶ −5.7 ± 4.4 *p* > 0.05
Kohns et al. [38], in 2020 RCT Michigan (USA)	LBP or Fibromyalgia ≥ 3 months N = 104 Patients recruited from the online registry: University of Michigan Health Research Volunteer Pool (umhealthresearch. org)	N = 51, M = 15, F = 35, Other = 1, Age: 44.35 ± 14.87 A single PPN session lasting 20–25 min, used a 3-min instructional video. Topics: pain, role of the brain in chronic pain, anatomy and physiology of the nervous system, perception of pain. At the end of the video, 5 self-assessment exercises to identify the presence of risk factors for central sensitization pain (compilation of body chart, assessment scales for risk factors, identification of personality traits related to chronic pain, identification of events that may affect chronic pain, Adverse Childhood Experience Scale). After a month and then after 10 months, participants were sent a link to a survey that contained follow-up measures.	N = 53, M = 11, F = 41, Other = 1 Age: 44.34 ± 14.69 Single session of self-assessment of health-related behaviors, lasting 20–25 min, used an educational video. Topics: 4 rules for a healthy lifestyle. at the end of the video 5 exercises to identify the habits of participants regarding: diet, exercise, sleep, hygiene, social relations. After a month and then after 10 months, participants were sent a link to a survey that contained follow-up measures.	*Brief Pain Inventory* (BPI) *Patient-Reported Outcomes Measurement Information System* (PROMIS) *Pain Stages of Change Questionnaire* (PSOCQ) *Pain Catastrophizing Scale* (PCS-13) *Tampa Scale of Kinesiophobia* (TSK-11) *Satisfaction with Life Scale* (SWLS) Baseline 1 month 10 months	BPI (severity) EX: 4.98 ± 1.54 ⟶ 4.03 ± 0.18 ⟶ 4.46 ± 0.23 C: 4.50 ± 1.73 ⟶ 4.60 ± 0.17 ⟶ 4.21 ± 0.23 *p* = 0.024 1 month *p* = 0.434 10 months BPI (interference) EX: 4.89 ± 2.44 ⟶ 3.91 ± 0.26 ⟶ 4.36 ± 0.29 C: 4.86 ± 2,64 ⟶ 4.71 ± 0.25 ⟶ 4.58 ± 0.28 *p* = 0.031 1 month *p* = 0.601 10 months PROMIS EX: 1.41 ± 0.77 ⟶ 1.34 ± 0.07 ⟶ 1.45 ± 0.08 C: 1.40 ± 0.85 ⟶ 1.47 ± 0.07 ⟶ 1.49 ± 0.08 *p* = 0.189 1 month *p* = 0.720 10 months PSOCQ EX: 8.28 ± 2.04 ⟶ 8.87 ± 0.19 ⟶ 8.64 ± 0.24 C: 8.02 ± 1.79 ⟶ 8.09 ± 0.19 ⟶ 8.39 ± 0.23 *p* = 0.005 1 month *p* = 0.447 10 months PCS-13 EX: 17.94 ± 10.94 ⟶ 15.44 ± 1.09 ⟶ 14.26 ± 1.03 C: 18.47 ± 12.15 ⟶ 17.61 ± 1.07 ⟶ 14.84 ± 1.01 *p* = 0.162 1 month *p* = 0.687 10 months TSK-11 EX: 24.55 ± 7.63 ⟶ 22.55 ± 0.54 ⟶ 23.77 ± 0.56 C: 24.13 ± 5.72 ⟶ 24.28 ± 0.53 ⟶ 24.16 ± 0.55 *p* = 0.082 1 month *p* = 0.623 10 months SWLS EX: 16.63 ± 7.08 ⟶ 17.71 ± 0.57 ⟶ 18.36 ± 0.70 C: 17.58 ± 8.43 ⟶ 17.30 ± 0.56 ⟶ 17.73 ± 0.69 *p* = 0.618 1 month *p* = 0.524 10 months
Malfliet et al. [39], in 2018 RCT Belgium	Chronic Spinal Pain (CSP) > 3 months N = 120 Age: 18–65 Patients recruited from two university medical centers through flyers, advertisements, social media. Patients divided into groups according to CSI levels.	N = 60, divided into: G1 = 24 (high CSI), M = 7, F = 17 Age: 36.58 ± 11.03 G2 = 36 (low CSI), M = 15, F = 21 Age: 40.47 ± 12.49 3 sessions in 2 weeks - First group session (max 6 people) with PNE education, used power point presentation (30–60 min). delivered a brochure to read at home. - Second online session with 3 videos and a final questionnaire. Topic for first and second session: anatomy and physiology of the nervous system, pain, factors influencing pain and central and peripheral sensitization. - Third individual session, 30 min of conversation one-by-one, with attention to the patient’s personal needs and discussion of the contents of previous sessions. Professional in charge: physiotherapist with experience on CSP	N = 60, divided into: G3 = 30 (high CSI) M = 8, F = 22 Age: 40.13 ± 14.91 G4 = 30 (low CSI) M = 17, F = 13 Age: 42.10 ± 11.10 Three sessions in 2 weeks - First group session (max 6 people) with education based on Neck/Back school guidelines, used power point presentation (30–60 min), delivered a brochure to read at home. - Second online session with 3 videos and a final questionnaire. Topics for first and second session: course and mechanical causes of pain, anatomy and physiology of the musculoskeletal apparatus, ergonomics and exercises (omitted information on the nervous system). - Third individual session, 30 min of conversation one-by-one, with attention to the patient’s personal needs and discussion of the contents of previous sessions. Professional in charge: physiotherapist with experience on CSP	*Pain Disability Index* (PDI) *Pain Catastrophizing Scale* (PCS-13) *Tampa Scale of Kinesiophobia* (TSK-17) *Revised Illness Perception Questionnaire* (IPQ-r) *Pain Vigilance Awareness Questionnaire* (PVAQ) Baseline 2 weeks (post-education)	PDI EX1: 30.09 ± 2.70 ⟶ 27.65 ± 2.41 C3: 26.03 ± 2.16 ⟶ 28.53 ± 2.11 EX2:16.25 ± 2.16 ⟶ 16.58 ± 1.93 C4: 17.13 ± 2.36 ⟶ 16.93 ± 2.11 *p* < 0.001 PCS-13 (Rumination) EX1: 8.96 ± 0.79 ⟶ 7.21 ± 0.78 C3: 7.63 ± 0.70 ⟶ 6.67 ± 0.70 EX2: 4.89 ± 0.64 ⟶ 5.33 ± 0.64 C4: 5.37 ± 0.70 ⟶ 4.77 ± 0.70 *p* < 0.001 PCS-13 (Magnification) EX1: 3.75 ± 0.43 ⟶ 2.83 ± 0.43 C3: 3.23 ± 0.38 ⟶ 2.87 ± 0.39 EX2: 1.58 ± 0.35 ⟶ 2.36 ± 0.35 C4: 2.27 ± 0.38 ⟶ 1.93 ± 0.39 *p* < 0.001 PCS-13 (Helplessness) EX1: 9.63 ± 0.98 ⟶ 7.96 ± 0.99 C3: 8.73 ± 0.88 ⟶ 7.47 ± 0.88 EX2: 6.19 ± 0.80 ⟶ 5.92 ± 0.80 C4: 6.47 ± 0.88 ⟶ 5.87 ± 0.88 *p* < 0.001 PCS-13 (total score) EX1: 22.33 ± 1.93 ⟶ 18.00 ± 1.93 C3: 19.60 ± 1.73 ⟶ 17.00 ± 1.72 EX2: 12.37 ± 1.58 ⟶ 13.61 ± 1.57 C4: 14.10 ± 1.73 ⟶ 12.57 ± 1.72 *p* > 0.05 TSK-17 EX1: 37.00 ± 1.39 ⟶ 32.25 ± 1.43 C3: 37.97 ± 1.24 ⟶ 36.53 ± 1.28 EX2: 32.61 ± 1.13 ⟶ 20.03 ± 1.17 C4: 35.47 ± 1.24 ⟶ 34.93 ± 1.28 *p* < 0.001 IPQ-r (Acute/Chronic timeline) EX1: 24.63 ± 0.80 ⟶ 20.58 ± 0.93 C3: 23.13 ± 0.71 ⟶ 22.17 ± 0.83 EX2: 23.33 ± 0.65 ⟶ 19.47 ± 0.76 C4: 23.33 ± 0.71 ⟶ 21.00 ± 0.83 IPQ-r (Consequence) EX1: 19.50 ± 0.89 ⟶ 16.96 ± 0.84 C3: 18.00 ± 0.79 ⟶ 17.90 ± 0.75 EX2: 14.53 ± 0.72 ⟶ 12.94 ± 0.69 C4: 15.30 ± 0.79 ⟶ 15.00 ± 0.75 IPQ-r (Personal Control) EX1: 19.33 ± 0.83 ⟶ 22.50 ± 0.63 C3: 19.43 ± 0.74 ⟶ 21.87 ± 0.56 EX2: 20.75 ± 0.67 ⟶ 22.39 ± 0.51 C4: 22.27 ± 0.74 ⟶ 22.43 ± 0.56 IPQ-r (Treatment control) EX1: 16.42 ± 0.56 ⟶ 17.75 ± 0.45 C3: 16.63 ± 0.50 ⟶ 17.07 ± 0.40 EX2: 17.11 ± 0.46 ⟶ 18.03 ± 0.37 C4: 17.83 ± 0.50 ⟶ 18.30 ± 0.40 IPQ-r (Illness Coherence) EX1: 16.88 ± 0.50 ⟶ 18.17 ± 0.52 C3: 15.63 ± 0.45 ⟶ 16.30 ± 0.46 EX2: 17.17 ± 0.41 ⟶ 17.19 ± 0.42 C4: 17.27 ± 0.45 ⟶ 18.07 ± 0.46 IPQ-r (Timeline Cyclical) EX1: 12.83 ± 0.65 ⟶ 14.17 ± 0.67 C3: 12.53 ± 0.58 ⟶ 12.77 ± 0.60 EX2: 13.28 ± 0.53 ⟶ 14.42 ± 0.54 C4: 13.80 ± 0.58 ⟶ 13.13 ± 0.60 IPQ-r (Emotional representations) EX1: 17.21 ± 0.93 ⟶ 17.04 ± 0.95 C3: 15.67 ± 0.83 ⟶ 16.00 ± 0.85 EX2: 13.42 ± 0.76 ⟶ 14.19 ± 0.77 C4: 13.40 ± 0.83 ⟶ 13.93 ± 0.85 *p* < 0.001 PVAQ EX1: 40.88 ± 2.49 ⟶ 35.38 ± 2.50 C3: 36.10 ± 2.22 ⟶ 34.97 ± 2.23 EX2: 34.25 ± 2.03 ⟶ 32.61 ± 2.04 C4: 35.43 ± 2.22 ⟶ 32.60 ± 2.23 *p* > 0.05
Meeus et al. [40], in 2010 RCT Belgium	Chronic Fatigue Syndrome (CFS) according to the criteria of: Centers for Disease Control and Prevention for CFS (1994) [51] N = 48 Age: 18–65 Patients recruited from the medical records available at the specialized university clinic in Brussels.	N = 24, M = 2, F = 22 Age: 38.3 ± 10.6 PNE: execution of an educational session based on the contents of the book “Explain Pain” with the help of images and examples. Topics retracted in another individual interactive session.	N = 24, M = 6, F = 18 Age: 42.3 ± 10.2 Education to self-management of ADL. Performing an educational session to promote a balance between rest and activity in patients, in order to avoid exacerbation of symptoms.	*Pain Coping Inventory* (PCI) *Pain Catastrophizing Scale* (PCS-13) *Tampa Scale of Kinesiophobia* (TSK-CFS) Pre-session (Baseline) Post-session	PCI (transforming) EX: 2.48 ± 0.71 ⟶ 2.30 ± 0.64 C: 2.23 ± 0.80 ⟶ 2.09 ± 0.62 *p* > 0.01 PCI (distraction) EX: 2.22 ± 0.54 ⟶ 2.32 ± 0.62 C: 2.51 ± 0.60 ⟶ 2.41 ± 0.52 *p* = 0.021 PCI (reducing demands) EX: 2.60 ± 0.59 ⟶ 2.29 ± 0.70 C: 2.71 ± 0.76 ⟶ 2.60 ± 0.85 *p* > 0.01 PCI (retreating) EX: 2.27 ± 0.80 ⟶ 2.21 ± 0.73 C: 2.53 ± 0.57 ⟶ 2.53 ± 0.57 *p* > 0.01 PCI (worrying) EX: 2.11 ± 0.41 ⟶ 1.85 ± 0.40 C: 2.09 ± 0.67 ⟶ 2.02 ± 0.67 *p* = 0.011 PCI (resting) EX: 2.41 ± 0.59 ⟶ 2.15 ± 0.59 C: 2.60 ± 0.64 ⟶ 2.56 ± 0.69 *p* > 0.01 PCS-13 (helplessness) EX: 8.54 ± 5.39 ⟶ 6.63 ± 4.70 C: 9.96 ± 6.77 ⟶ 9.29 ± 6.93 *p* > 0.01 PCS-13 (magnification) EX: 2.38 ± 1.93 ⟶ 1.67 ± 1.55 C: 4.00 ± 3.51 ⟶ 3.75 ± 4.19 *p* > 0.01 PCS-13 (ruminating) EX: 7.29 ± 3.86 ⟶ 5.71 ± 3.46 C: 7.83 ± 4.59 ⟶ 7.67 ± 4.91 *p* = 0.009 TSK-CFS EX: 39.17 ± 9.52 ⟶ 33.21 ± 6.58 C: 39.71 ± 7.15 ⟶ 37.42 ± 8.15 *p* > 0.05
Orhan et al. [41], in 2021 RCT Belgium	CLBP > 3 months N = 29 Age: 18–65 Patients recruited at a private medical centre in Ghent (BE).	N = 15, M = 4, F = 11 Age: 55.0 (47.0–59.0) 2 educational sessions with PNE. approach developed during a study “Delphi modified”(Orhan et al., 2019). The same contents as the control group, but with adaptations based on the culture of the participants or gender. Topics: difference between chronic and acute pain, role of pain, genesis of pain, factors that promote central awareness and treatment strategies. use of verbal information, images, metaphors based on previous research and books including: “*Explain Pain*” (Butler and Moseley, 2003) and “Pijneducatie: Een Praktische Handleiding voor (Para) medici” (Van Wilgen e Nijs, 2010). Professional in charge: first author instructed by 2 physiotherapists experts in PNE.	N = 14, M = 4 F = 10 Age: 55.0 (45.0–60.2) Two educational sessions with standard PNE (same as the intervention group). Standard content translated into Turkish by 2 independent native Turkish-speaking translators. Topics: difference between chronic and acute pain, role of pain, genesis of pain, factors that promote central awareness and treatment strategies. use of verbal information, images, metaphors based on previous research and books including: “*Explain Pain*” (Butler and Moseley, 2003) and “Pijneducatie: Een Praktische Handleiding voor (Para) medici” (Van Wilgen e Nijs, 2010). Professional in charge: first author instructed by 2 physiotherapists experts in PNE.	*Numerical Rating Scale* (NRS) *Roland Morris Disability* Questionnaire (RMDQ) *Pain Beliefs Questionnaire* (PBQ) *Pain Catastrophizing Scale* (PCS-13) *Tampa Scale of Kinesiophobia* (TSK-17) Baseline 1 weeks 4 weeks	NRS EX: 6.50 ± 1.80 ⟶ 5.80 ± 2.07 ⟶ 5.86 ± 2.35 C: 6.85 ± 2.21 ⟶ 5.85 ± 2.24 ⟶ 6.00 ± 2.48 *p* > 0.05 RMDQ EX: 16.66 ± 4.32 ⟶ 15.40 ± 4.82 ⟶ 15.60 ± 6.12 C: 16.21 ± 4.62 ⟶ 14.42 ± 5.98 ⟶ 13.07 ± 5.91 *p* > 0.05 PBQ (organic score) EX: 4.23 ± 0.93 ⟶ 3.67 ± 0.99 ⟶ 3.87 ± 0.65 C: 4.02 ± 0.75 ⟶ 3.64 ± 0.70 ⟶ 3.50 ± 0.92 *p* > 0.05 PBQ (psychological score) EX: 4.10 ± 1.22 ⟶ 4.86 ± 1.22 ⟶ 4.36 ± 1.14 C: 4.35 ± 0.93 ⟶ 4.51 ± 1.04 ⟶ 4.12 ± 1.07 *p* > 0.05 PCS-13 EX: 29.40 ± 10.68 ⟶ 25.93 ± 11.21 ⟶ 24.80 ± 11.21 C: 24.14 ± 10.86 ⟶ 19.35 ± 10.77 ⟶ 19.00 ± 11.08 *p* > 0.05 TSK-17 EX: 45.33 ± 5.17 ⟶ 42.73 ± 5.37 ⟶ 43.26 ± 6.06 C: 43.64 ± 3.65 ⟶ 41.71 ± 3.45 ⟶ 40.50 ± 3.87 *p* > 0.05
Pires et al. [42], in 2015 RCT Portugal	CLBP > 3 months N = 62 Age: 18–65 Patients recruited from the waiting list of a Portuguese clinic	N = 30, M = 10, F = 20 Age: 50.9 ± 6.2 1. PNE: 2 group session of 90 min each, for educational sessions were used as help metaphors and images shown before the program of the exercise. Topics: neurophysiology of pain, central sensitization, psychosocial factors 2. Aquatic exercise program (same as control)	N = 32, M = 12, F = 20 Age: 51.0 ± 6.3 1. Aquatic exercise program: Group exercises, 2 sessions per week for 6 weeks, each lasting 30–50 min with warm-up, activity, cooling-off.	*Visual Analogue Scale* (VAS) *Quebec Back Pain Disability Scale* (QBPDS-PT) *Tampa Scale of Kinesiophobia* (TSK-13) Baseline 6 weeks 3 months	VAS EX: 43.4 ± 23 ⟶ 20.6 ± 19 ⟶ 18.0 ± 19 C: 42.4 ± 21.2 ⟶ 27.6 ± 17.2 ⟶ 35.8 ± 28 *p* = 0.14 6 weeks *p* < 0.05 3 months QBPDS-PT EX: 32.3 ± 14 ⟶ 21.2 ± 15.8 ⟶ 19.2 ± 14.8 C: 28.1 ± 13.6 ⟶ 20.4 ± 12.3 ⟶ 25.9 ± 15.7 *p* > 0.05 TSK-13 EX: 28.6 ± 6 ⟶ 25.2 ± 4.7 ⟶ 23.2 ± 6.3 C: 29.1 ± 5.6 ⟶ 27.5 ± 6.2 ⟶ 26.5 ± 7.9 *p* > 0.05
Rabiei et al. [43], in 2021 RCT Iran	CLBP > 3 months N = 73 Age: 30–60 Patients recruited by physiotherapists through leaflets exposed in rehab clinics.	N = 37, M = 16, F = 21 Age: 42.46 ± 9.7 1. PNE: 3 educational sessions, each lasting 30–60 min. Frontal sessions, with the help of diagrams and drawings Professional in charge: Persian native physiotherapist, trained in PNE and MCE programs. 2. MCE, 2 sessions a week for 8 weeks. In the first session, each patient was assessed by the physiotherapist. The training included: sensory-motor control training, proprioception and recruitment (diaphragm, multifidus, transverse abdomen, pelvic floor) in progression. Initially proposed static exercises, then dynamic exercises related to functional activities. Objective: coordination, posture, stability of the spine, encouraging the resumption of activities feared by the patient in daily life. Professional in charge: Persian native physiotherapist, trained in PNE and MCE programs.	N = 36, M = 18, F = 18 Age: 44.19 ± 8.79 1. Group-based exercise (GE) program. Proposed sessions 2 times a week for 8 weeks, each session lasting 60 min (10 min group warm-up, 45 min muscle strengthening exercises, 5 min light exercises). For each exercise: 3 series of 10 repetitions each. The program was modified in relation to patient tolerance. Professional in charge: physiotherapist not involved in the intervention group	*Visual Analogue Scale* (VAS) *Roland Morris Disability* Questionnaire (RMDQ) Fear Avoidance Beliefs Questionnaire (FABQ) *Pain Self Efficacy Questionnaire* (PSEQ) Baseline 8 weeks	VAS EX: 6.45 ± 1.21 ⟶ 3.79 ± 1.02 C: 6.36 ± 1.14 ⟶ 4.91 ± 1.67 *p* = 0.041 RMDQ EX: 14.6 ± 1.55 ⟶ 7.94 ± 2.17 C: 15.0 ± 2.14 ⟶ 9.50 ± 3.25 *p* = 0.021 FABQ-W EX: 24.2 ± 10.4 ⟶ 11.5 ± 6.41 C: 21.6 ± 8.02 ⟶ 14.9 ± 6.43 *p* = 0.819 FABQ-PA EX: 17.2 ± 4.25 ⟶ 8.24 ± 3.72 C: 15.7 ± 5.17 ⟶ 10.2 ± 4.15 *p* = 0.803 PSEQ EX: 26.6 ± 9.53 ⟶ 43.9 ± 11.6 C: 29.5 ± 10.9 ⟶ 38.9 ± 12.0 *p* = 0.661
Ryan et al. [44], in 2010 RCT UK	CLBP > 3 months N = 38 Age: 18–65 Patients recruited from 5 different physical therapy departments	N = 18, M = 7, F = 11 Age: 45.2 ± 9.5 1.PNE: a single educational session of 2 h and 30 min on the biology of pain. oral session, with the help of diagrams, drawings and delivery of the book: “The back book”	N = 20, M = 6, F = 14 Age: 45.2 ± 11.9 1.PNE (same as intervention group) 2. Circuit aerobics exercises associated with core stability exercises (“Back to fitness exercise classes”). each session divided into: 10 min warm-up, 20–30 min aerobic phase, 10–15 min cool-down). 6 exercise classes, one class per week for 6 weeks.	*Roland Morris Disability* Questionnaire (RMDQ) *Numerical Rating Scale* (NRS) *Tampa Scale of Kinesiophobia* (TSK-13) *Pain Self Efficacy Questionnaire* (PSEQ) Baseline 8 weeks 3 months	RMDQ EX: 10.8 ± 5.2 ⟶ 3.3 ± 3.0 ⟶ 4.3 ± 4.2 C: 9.4 ± 4.2 ⟶ 5.6 ± 3.9 ⟶ 6.4 ± 5.1 *p* > 0.05 NRS EX: 39.3 ± 26.2 ⟶ 8.4 ± 7.5 ⟶ 22.6 ± 30.8 C: 28.1 ± 20.4 ⟶ 23.9 ± 23.3 ⟶ 19.1 ± 18.9 *p* = 0.025 8 weeks *p* > 0.05 3 months TSK-13 EX: 28.4 ± 8.2 ⟶ 21.3 ± 6.5 ⟶ 23.7 ± 6.6 C: 25.8 ± 7.4 ⟶ 21.9 ± 8.2 ⟶ 21.5 ± 7.5 *p* > 0.05 PSEQ EX: 41.9 ± 12.5 ⟶ 55.1 ± 4.7 ⟶ 49.5 ± 9.8 C: 50.0 ± 11.4 ⟶ 48.8 ± 12.2 ⟶ 49.5 ± 13.1 *p* = 0.024 8 weeks *p* > 0.05 3 months
Saracoglu et al. [45], in 2020 RCT Turkey	CLBP > 6 months N = 69 Age: 18–65 Patients recruited by the physical therapy department of Kutahya University Hospital	The participants were divided into 3 groups. Group 1 ⟶ N = 20, M = 9, F = 12 1. PNE: according to the method recommended by Louw, Nijs and Puentedura (2017). 4 individual educational sessions, one per week, lasting about 40–45 min, after the manual therapy session. Topics: peripheral pain, allodynia, central and peripheral sensitization, hyperalgesia, neuroplasticity, stress, fear, catastrophization, immunological response, how to deal with pain, role of physical exercise and manual therapy. In support, used tools such as: metaphors, presentations, images. Professional in charge: physiotherapist trained at the International Spine and Pain Institute 2. Manual Therapy: same group 2 modes. 3. Home exercise program: same group 2 and group 3 modes.	Group 2 ⟶ N = 19, M = 11, F = 10 Group 3 ⟶ N = 18, M = 10, F = 10 Two control groups: Group 2: 1. Manual therapy: personalized treatment, 2 sessions per week, for 4 weeks, each lasting 30 min. Use different techniques with variable speed, range, direction of force application and patient position. Professional in charge: same experienced physiotherapist of PNE, with University Master and 10 years of experience in manual therapy. 2. Home exercise program, developed by Koumantakis, Watson and Oldham (2005). Objectives: to increase strength and flexibility of the trunk muscles, buttocks, quadriceps and ischiocrural. the program includes: stretching, heating, reinforcement. indication: 10 repetitions, 3 times a day for 4 weeks. Professional in charge: physiotherapist. Group 3: only home exercise program, with the same group 1 and group 2 modes.	*Numerical Pain Rating Scale* (NPRS) *Oswestry Disability Index* (ODI) *Tampa Scale of Kinesiophobia* (TSK-17) Baseline 4 weeks 12 weeks	NPRS G1: 7.29 ± 1.41 ⟶ 3.05 ± 1.50 ⟶ 2.09 ± 1.64 G2: 7.58 ± 1.47 ⟶ 4.42 ± 1.78 ⟶ 4.52 ± 1.84 C: 7.32 ± 1.48 ⟶ 5.89 ± 2.03 ⟶ 5.47 ± 1.95 *p* (G1 vs. G2) = 0.01 *p* (G1 vs. G3) < 0.001 ODI G1: 34.45 ± 7.39 ⟶ 22.80 ± 6.77 ⟶ 19.90 ± 5.72 G2: 32.00 ± 6.87 ⟶ 25.00 ± 7.88 ⟶ 25.89 ± 7.37 C: 34.74 ± 8.55 ⟶ 31.77 ± 9.27 ⟶ 32.33 ± 8.49 *p* (G1 vs. G2) = 0.67 *p* (G1 vs. G3) < 0.001 TSK-17 G1: 44.35 ± 4.30 ⟶ 35.55 ± 5.75 ⟶ 35.19 ± 3.99 G2: 45.10 ± 4.45 ⟶ 41.63 ± 5.23 ⟶ 42.21 ± 5.04 C: 45.55 ± 4.10 ⟶ 44.94 ± 4.70 ⟶ 44.88 ± 5.10 *p* (G1 vs. G2) < 0.001 *p* (G1 vs. G3) < 0.001
Serrat et al. [46], in 2020 RCT Spain	Fibromyalgia according to American College of Reumatology criteria (ACR) [38] N = 169 Age: ≥18 Patients recruited at the university hospital in Val d’Hebron (UHVH) by a specialized physiotherapist.	N = 84, M = 2, F = 82 Age: 54.12 ± 8.62 1. NAT-FM: 2-h session per week for 12 weeks. Outdoor sessions, different approaches were used: - PNE: (20 min) reconceptualization of pain, anatomy and neurophysiology of the nervous system, modulation of pain, influencing factors, types of pain, neuromatrix theory, neuroplasticity. - Therapeutic exercise and activities in nature: (40 min) hiking, yoga, nordic walking, photography, Shirin yoku. - Cognitive behavioral therapy (CBT): (20 min) - Mindfulness Training (MT): (20 min) 2. TAU: same as control group Professional in charge: physiotherapist, sports psychologist and technician of the CSSU of the Val d’Hebron university hospital, prepared to conduct this treatment, supported by a patient with fibromyalgia who brought his testimony in groups to motivate participants in compliance, after successfully completing the program FIBROWALK	N = 85, M = 0, F = 85 Age: 53.15 ± 9.06 1.TAU: 2-h session per week for 12 weeks. Includes: basic disease education, aerobic exercises, personalized pharmacological treatment. integration of the FIBROWALK protocol.	*Fibromyalgia Impact Questionnaire* (FIQ) *Visual Analogue Scale* (VAS) *Hospital Anxiety and Depression Scale* (HADS) *Short Form Health Survey 36* (SF-36) *Rosenberg Self-Esteem Scale* (RSES) *Tampa Scale of Kinesiophobia* (TSK-11) *Pain Catastrophizing Scale* (PCS-13) Baseline 6 weeks 12 weeks	FIQR EX: 73.07 ± 13.79 ⟶ 58.78 ± 18.70 ⟶ 50.69 ± 18.05 C: 73.21 ± 14.72 ⟶ 69.68 ± 13.36 ⟶ 69.18 ± 17.88 *p* < 0.001 VAS pain EX: 7.74 ± 1.52 ⟶ 6.78 ± 1.99 ⟶ 5.60 ± 1.98 C: 7.80 ± 1.61 ⟶ 7.52 ± 1.59 ⟶ 7.47 ± 1.79 *p* = 0.003 6 weeks *p* < 0.001 12 weeks VAS fatigue EX: 7.61 ± 1.89 ⟶ 5.98 ± 2.10 ⟶ 5.58 ± 2.00 C: 7.76 ± 1.91 ⟶ 7.32 ± 2.09 ⟶ 7.08 ± 2.34 *p* = 0.002 6 weeks *p* < 0.001 12 weeks HADS-A EX: 13.95 ± 3.80 ⟶ 11.03 ± 4.25 ⟶ 10.16 ± 4.19 C: 13.13 ± 4.22 ⟶ 12.35 ± 4.07 ⟶ 12.68 ± 4.63 *p* < 0.001 HADS-D EX: 11.27 ± 3.71 ⟶ 9.66 ± 4.47 ⟶ 8.18 ± 4.42 C: 11.49 ± 4.64 ⟶ 11.22 ± 5.02 ⟶ 11.67 ± 5.18 *p* = 0.027 6 weeks *p* < 0.001 12 weeks SF-36 EX: 27.03 ± 18.85 ⟶ 35.09 ± 20.47 ⟶ 43.42 ± 20.92 C: 26.04 ± 18.11 ⟶ 28.24 ± 17.38 ⟶ 25.07 ± 15.86 *p* = 0.017 6 weeks *p* < 0.001 12 weeks RSES EX: 16.03 ± 3.36 ⟶ 16.60 ± 2.70 ⟶ 16.53 ± 2.25 C: 15.41 ± 3.57 ⟶ 15.48 ± 2.57 ⟶ 16.25 ± 3.45 *p* > 0.05 TSK-11 EX: 29.23 ± 7.40 ⟶ 21.36 ± 6.83 ⟶ 17.95 ± 4.97 C: 29.92 ± 7.58 ⟶ 25.59 ± 6.46 ⟶ 28 ± 7.44 *p* < 0.001 PCS-13 EX: 27.04 ± 11.33 ⟶ 17.83 ± 9.56 ⟶ 13.53 ± 8.87 C: 27.72 ± 12.65 ⟶ 26.72 ± 13.25 ⟶ 27.49 ± 13.35 *p* < 0.001
Téllez-Garcia et al. [47], in 2015 RCT Spain	Non-specific CLBP ≥ 3 months N= 12 Age: 18–65 Patients with non-specific LBP recruited had gone to the doctor to request physiotherapy sessions	N = 6, M = 2, F = 4 Age: 36 ± 5 1. Dry needling (same as control group) 2. PNE: 2 individual educational sessions, one to one lasting 30 min each, once a week immediately after the 2° and 3° sessions of Dry needling. Topics: neurophysiology of pain, beliefs about pain. Used power point material based on the book “*Explain Pain*” by Butler and Moseley (2003) and other material to read at home.	N = 6, M = 2, F = 4 Age: 37 ± 13 Three dry needling sessions, once a week. run on active trigger point (gluteus medius, quadratus lumborum) with fast-in and fast-out techniques by Hong. Professional in charge: clinical expert	*Numerical Pain Rating Scale* (NPRS) *Oswestry Disability Index* (ODI) - *Roland Morris Disability* Questionnaire (RMDQ) *Tampa Scale of Kinesiophobia* (TSK-17) Baseline 4 weeks	NPRS EX: 5.0 ± 2.1 ⟶ 0.8 ± 1.0 C: 4.8 ± 3.1 ⟶ 1.2 ± 1.1 *p* > 0.05 ODI EX: 24.2 ± 9.7 ⟶ 4.7 ± 3.2 C: 30.0 ± 14.8 ⟶ 6.0 ± 5.1 *p* > 0.05 RMDQ EX: 10.3 ± 3.4 ⟶ 1.0 ± 1.1 C: 8.3 ± 1.2 ⟶ 2.2 ± 2.2 ± 0.8 *p* > 0.05 TSK-17 EX: 41.5 ± 6.2 ⟶ 23.8 ± 2.9 C: 43.3 ± 5.9 ⟶ 38.3 ± 5.1 *p* = 0.008 MCID > 8
Van Ittersum et al. [48], in 2014 RCT Belgium	Fibromyalgia according to American College of Reumatology criteria (ACR) [38] N = 105 Age: 18–65 Patients recruited from two centers specialized in chronic pain and cfs, in Belgium	N = 53, M = 6%, F = 94% Age: 47.6 ± 9.1 For both groups, a period of 6 weeks was considered for reading, learning and applying the contents of the brochures. To the intervention group proposed education through PNE with delivery of a 15 page brochure containing images and written information about: neurophysiology of pain, anatomy and physiology of the nervous system, based on the book “*Explain Pain*” by Butler and Moseley (2003). After 2 weeks, the physiotherapist called the patients to make sure they understood the contents of the brochure and answered any questions.	N = 52, M = 8%, F = 92% Age: 45.8 ± 9.8 To the control group, proposed a brochure of 15 pages containing relaxation exercises and instructions to perform them independently at home. Explanation of the relationship between pain and physical and psychological factors (Loeser model). Patients could choose one or more exercises on the 3 techniques proposed (Jacobson’s progressive muscle relaxation method, visualization method derived from meditation techniques, autogenous training described by Schultz). After 2 weeks, the physiotherapist called the patients same as intervention group.	*Fibromyalgia Impact Questionnaire* (FIQ) *Revised Illlness Perception Questionnaire* (IPQ-R_FM) *Pain Catastrophizing Scale* (PCS) Baseline 6 weeks 6 months	FIQ EX: 70.0 ± 14.8 ⟶ 69.3 ± 15.4 ⟶ 67.4 ± 15.5 C: 66.6 ± 14.8 ⟶ 65.1 ± 15.2 ⟶ 67.1 ± 15.1 *p* > 0.05 IPQ-R FM: IPQ-r (Acute/Chronic timeline) EX: 17.4 ± 2.8 ⟶ 18.0 ± 3.0 ⟶ 21.0 ± 5.0 C: 17.6 ± 2.3 ⟶ 18.5 ± 3.0 ⟶ 23.4 ± 4.5 IPQ-r (Timeline Cyclical) EX: 14.7 ± 3.9 ⟶ 14.3v3.7 ⟶ 14.8 ± 3.3 C: 14.1 ± 3.4 ⟶ 14.7 ± 3.5 ⟶ 14.3 ± 3.3 IPQ-r (Consequence) EX: 20.3 ± 3.9 ⟶ 20.4 ± 3.6 ⟶ 22.2 ± 3.7 C: 20.4 ± 3.5 ⟶ 21.0 ± 3.4 ⟶ 23.4 ± 3.9 IPQ-r (Personal Control) EX: 16.0 ± 3.4 ⟶ 16.9 ± 3.3 ⟶ 16.4 ± 3.9 C: 16.8 ± 2.2 ⟶ 15.2 ± 2.6 ⟶ 17.7 ± 3.7 IPQ-r (Treatment control) EX: 15.4 ± 2.4 ⟶ 15.1 ± 2.4 ⟶ 14.8 ± 3.1 C: 15.3 ± 2.1 ⟶ 15.2 ± 2.6 ⟶ 14.9 ± 3.1 IPQ-r (Emotional representations) EX: 16.6 ± 4.6 ⟶ 16.5 ± 5.0 ⟶ 17.1 ± 5.2 C: 15.1 ± 4.3 ⟶ 15.0 ± 4.0 ⟶ 16.1 ± 4.7 IPQ-r (Illness Coherence) EX 15.6 ± 4.0 ⟶ 15.1 ± 3.9 ⟶ 15.7 ± 3.3: C: 15.8 ± 3.6 ⟶ 15.4 ± 3.7 ⟶ 15.4 ± 3.3 IPQ-r (Psychologica attribution) EX: 17.7 ± 5.3 ⟶ 18.1 ± 5.3 ⟶ 17.8 ± 5.3 C: 17.0 ± 5.8 ⟶ 18.2 ± 5.1 ⟶ 17.4 ± 5.7 IPQ-r (Risk factor attribution) EX: 14.6 ± 4.0 ⟶ 15.1 ± 4.0 ⟶ 14.9 ± 4.2 C: 14.2 ± 4.0 ⟶ 14.9 ± 4.3 ⟶ 15.0 ± 4.1 IPQ-r (Immune attribution) EX: 8.7 ± 2.1 ⟶ 8.7 ± 1.9 ⟶ 8.7 ± 2.2 C: 8.5 ± 2.8 ⟶ 8.5 ± 2.6 ⟶ 8.8 ± 2.2 IPQ-r (Accident/chance) EX: 5.5 ± 1.8 ⟶ 5.3 ± 1.7 ⟶ 5.5 ± 1.6 C: 5.2 ± 2.0 ⟶ 5.3 ± 2.2 ⟶ 5.7 ± 2.0 IPQ-r (FM-specific attribution) EX: 23.4 ± 5.0 ⟶ 23.9 ± 4.5 ⟶ 23.5 ± 4.3 C: 23.5 ± 4.5 ⟶ 23.8 ± 4.5 ⟶ 23.6 ± 4.9 *p* > 0.05 PCS EX: 24.0 ± 11.9 ⟶ 24.1 ± 12.8 ⟶ 24.3 ± 13.4 C: 23.0 ± 12.1 ⟶ 21.7 ± 12.0 ⟶ 22.3 ± 12.8 *p* > 0.05
Van Oosterwjck et al. [49], in 2013 RCT Belgium	Fibromyalgia according to American College of Reumatology criteria (ACR) [38] N = 30 Age: 18–65 Patients recruited from private practices of internal medicine.	N = 15, M = 3, F = 12 Age: 45.8 ± 9.5 2 educational sessions of PNE lasting 30 min once a week, were proposed to the intervention group. The 1° oral session, the second by telephone call. Topics: neurophysiology of pain, nervous system plasticity, central sensitization, with reference to the book “*Explain Pain*” by Butler and Moseley (2003). As a support use power-point presentations, images, metaphors, examples. At the end of the first session, a booklet is given to read at home as a reinforcement.	N = 15, M = 1, F = 14 Age: 45.9 ± 11.5 2 individual educational sessions of pain self-management, lasting 30 min once a week, were proposed to the control group. The first oral session, the second by telephone call. Topics: self-management techniques for pain and symptoms. At the end of the 1° session, a booklet is given to read at home as a reinforcement.	*Fibromyalgia Impact Questionnaire* (FIQ) *Short Form Health Survey 36* (SF-36) *Pain Coping Inventory* (PCI) *Pain Catastrophizing Scale* (PCS) *Pain Vigilance and Awareness* (PVAQ) *Tampa Scale of Kinesiophobia* (TSK-17) Baseline 2 weeks 3 months	FIQ EX: 38.7 ± 10.7 ⟶ 34.9 ± 10.1 ⟶ 36.5 ± 9.9 C: 59.4 ± 12.9 ⟶ 58.7 ± 15.4 ⟶ 60.1 ± 10.5 *p* > 0.05 SF-36 “physical functioning” EX: 47.7 ± 22.7 ⟶ 51.0 ± 21.6 ⟶ 53.7 ± 21.8 C: 49.7 ± 17.9 ⟶ 45.7 ± 17.1 ⟶ 45.3 ± 12.3 *p* = 0.046 SF-36 “role limitations due to physical pain” EX: 18.3 ± 34.7 ⟶ 25.0 ± 35.4 ⟶ 28.3 ± 35.2 C: 13.3 ± 22.9 ⟶ 5.0 ± 10.4 ⟶ 15.0 ± 26.4 *p* > 0.05 SF-36 “bodily pain” EX: 37.1 ± 19.2 ⟶ 45.8 ± 25.8 ⟶ 42.5 ± 19.9 C: 40.3 ± 15.8 ⟶ 49.2 ± 20.2 ⟶ 52.4 ± 21.5 *p* > 0.05 SF-36 “general health perceptions” EX: 24.7 ± 10.6 ⟶ 32.8 ± 15.5 ⟶ 37.7 ± 15.5 C: 31.47 ± 12.8 ⟶ 33.3 ± 14.0 ⟶ 28.6 ± 12.8 *p* < 0.001 SF-36 “vitality” EX: 36.3 ± 17.8 ⟶ 35.7 ± 18.5 ⟶ 40.0 ± 21.0 C: 42.2 ± 14.3 ⟶ 38.5 ± 13.1 ⟶ 35.3 ± 13.7 *p* = 0.047 SF-36 “social functioning” EX: 63.1 ± 21.5 ⟶ 63.8 ± 27.0 ⟶ 58.1 ± 27.4 C: 48.4 ± 17.2 ⟶ 54.9 ± 20.2 ⟶ 61.2 ± 15.9 *p* > 0.05 SF-36 “role limitations due to emotional problems” EX: 71.1 ± 45.2 ⟶ 60.5 ± 43.1 ⟶ 59.9 ± 47.5 C: 42.2 ± 14.3 ⟶ 38.5 ± 13.1 ⟶ 35.3 ± 13.7 *p* > 0.05 SF-36 “mental health” EX: 60.8 ± 17.3 ⟶ 61.9 ± 22.4 ⟶ 66.7 ± 17.5 C: 62.0 ± 19.5 ⟶ 60.1 ± 20.8 ⟶ 48.5 ± 18.3 *p* < 0.001 PCI “transformation” EX: 2.1 ± 0.5 ⟶ 1.8 ± 0.6 ⟶ 1.9 ± 0.6 C: 1.7 ± 0.6 ⟶ 1.9 ± 0.7 ⟶ 1.9 ± 0.7 *p* > 0.05 PCI “distraction” EX: 2.2 ± 0.6 ⟶ 2.1 ± 0.8 ⟶ 2.1 ± 0.8 C: 2.1 ± 0.6 ⟶ 1.9 ± 0.6 ⟶ 2.0 ± 0.7 *p* > 0.05 PCI “reducing demands” EX: 2.2 ± 0.6 ⟶ 2.3 ± 0.8 ⟶ 2.2 ± 0.9 C: 2.0 ± 0.8 ⟶ 2.0 ± 0.9 ⟶ 2.1 ± 0.8 *p* > 0.05 PCI “worrying” EX: 1.7 ± 0.6 ⟶ 1.5 ± 0.5 ⟶ 1.5 ± 0.5 C: 1.7 ± 0.6 ⟶ 1.7 ± 0.7 ⟶ 1.6 ± 0.5 *p* > 0.05 PCI “retreating” EX: 1.7 ± 0.7 ⟶ 1.6 ± 0.6 ⟶ 1.7 ± 0.6 C: 1.9 ± 0.6 ⟶ 1.9 ± 0.7 ⟶ 1.8 ± 0.6 *p* > 0.05 PCI “resting” EX: 2.1 ± 0.4 ⟶ 1.9 ± 0.6 ⟶ 2.0 ± 0.8 C: 1.9 ± 0.7 ⟶ 1.9 ± 0.7 ⟶ 2.1 ± 0.6 *p* > 0.05 PCS “helplessness” EX: 9.5 ± 5.3 ⟶ 6.7 ± 6.0 ⟶ 6.7 ± 5.8 C: 11.1 ± 5.8 ⟶ 10.5 ± 5.6 ⟶ 9.6 ± 5.5 *p* > 0.05 PCS “magnification” EX: 2.8 ± 2.4 ⟶ 2.2 ± 2.5 ⟶ 1.7 ± 2.1 C: 3.0 ± 2.5 ⟶ 3.2 ± 2.1 ⟶ 3.3 ± 2.7 *p* > 0.05 PCS “rumination” EX: 7.1 ± 5.0 ⟶ 6.1 ± 3.8 ⟶ 5.1 ± 4.2 C: 7.7 ± 2.7 ⟶ 7.5 ± 3.3 ⟶ 7.5 ± 4.0 *p* > 0.05 PCS total EX: 19.5 ± 11.8 ⟶ 14.9 ± 11.6 ⟶ 13.3 ± 11.6 C: 21.9 ± 9.9 ⟶ 20.5 ± 10.2 ⟶ 20.4 ± 12.3 *p* > 0.05 PVAQ EX: 35.3 ± 14.5 ⟶ 34.4 ± 11.7 ⟶ 32.2 ± 13.7 C: 39.7 ± 12.6 ⟶ 42.6 ± 17.1 ⟶ 40.3 ± 14.4 *p* > 0.05 TSK-17 EX: 38.7 ± 10.7 ⟶ 34.9 ± 10.1 ⟶ 36.5 ± 9.9 C: 40.7 ± 8.4 ⟶ 39.8 ± 7.1 ⟶ 39.9 ± 8.2 *p* > 0.05

**Table 2 ijerph-20-04098-t002:** Drop-outs and lost to follow-up in the included studies.

Study	Drop-Outs			Lost to Follow-Up		
	Experimental Group	Control Group		Experimental Group	Control Group	
Barrenengoa-Cuadra et al. [35], in 2021	0	1 at baseline		2 at 12 months	2 at 12 months	
Bodes et al. [36], in 2018	0	0		0	0	
Gül et al. [37], in 2021	0	0		0	0	
Kohns et al. [38], in 2020	0	0		10 at 10 months	6 at 10 months	
Malfliet et al. [39], in 2018	0	0		5 at 2 weeks	4 at 2 weeks	
Meeus et al. [40], in 2010	2 at 30 min	0		0	0	
Orhan et al. [41], in 2021	4 at 1 week	4 at 1 week		0	0	
Pires et al. [42], in 2015	1 at 6 weeks	5 at 6 weeks		0	1 at 3 months	
Rabiei et al. [43], in 2021	3	4		0	0	
Ryan et al. [44], in 2010	2 at 8 weeks	2 at 8 weeks		3 at 3 months	4 at 3 months	
Saracoglu et al. [45], in 2020	0		0	3 at 12 weeks	4 at 12 weeks	5 at 12 weeks
Serrat et al. [46], in 2020	0	0		10 at 12 weeks	0	
Téllez-Garcia et al. [47], in 2015	0	0		0	0	
Van Ittersum et al. [48], in 2014	32 at 6 weeks of follow-up			14 at 6 months of follow-up		
Van Oosterwjck et al. [49], in 2013	0	0		3 at 3 months	1 at 3 months	

**Table 3 ijerph-20-04098-t003:** Outcome and outcome measures in the included studies.

Outcome Measures	Study in Which Outcome Measures Have Been Addressed
BPI—Brief Pain Inventory	Barrenengoa-Cuadra et al. (2021) [35], Kohns et al. (2020) [38]
FABQ—Fear Avoidance Beliefs Questionnaire	Rabiei et al. (2021) [43]
FIQ—Fibromyalgia Impact Questionnaire	Barrenengoa-Cuadra et al. (2021) [35], Van Ittersum et al. (2014) [48], Van Oosterwjck et al. (2013) [49]
FIQR—Fibromyalgia Impact Questionnaire	Serrat et al. (2020) [46]
HADS—Hospital Anxiety and Depression Scale	Barrenengoa-Cuadra et al. (2021) [35], Serrat et al. (2020) [46]
HAQ—Health Assessment Questionnaire	Barrenengoa-Cuadra et al. (2021) [35]
IPQ–r Illness Perception Questionnaire	Malfliet et al. (2018) [39]
IPQ-R FM—Revised Illness Perception Questionnaire	Van Ittersum et al. (2014) [48]
NPRS—Numerical Pain Rating Scale	Bodes et al. (2018) [36], Saracoglu et al. (2020) [45], Téllez-Garcia et al. (2015) [47]
NRS—Numerical Rating Scale	Orhan et al. (2021) [41], Ryan et al. (2010) [44]
ODI—Oswestry Disability Index	Saracoglu et al. (2020) [45], Téllez-Garcia et al. (2015) [47]
PBQ—Personality Belief Questionnaire	Orhan et al. (2021) [41]
PCI—Pain Coping Inventory	Meeus et al. (2010) [40], Van Oosterwjck et al. (2013) [49]
PCS—Pain Catastrophizing Scale	Barrenengoa-Cuadra et al. (2021) [35], Bodes et al. (2018) [36], Kohns et al. (2020) [38], Malfliet et al. (2018) [39], Meeus et al. (2010) [40], Orhan et al. (2021) [41], Serrat et al. (2020) [46], Van Ittersum et al. (2014) [48], Van Oosterwjck et al. (2013) [49]
PDI—Pain Disability Index	Malfliet et al. (2018) [39]
PROMIS—Patient-Reported Outcomes Measurement Information System	Kohns et al. (2020) [38]
PSEQ—Pain Self-Efficacy Questionnaire	Rabiei et al. (2021) [43], Ryan et al. (2010) [44]
PSOCQ—Pain Stages of Change Questionnaire	Kohns et al. (2020) [38]
PVAQ—Pain Vigilance and Awareness Questionnaire	Malfliet et al. (2018) [39], Van Oosterwjck et al. (2013) [36]
QBPDS-PT—Quebec Back Pain Disability Scale	Pires et al. (2015) [42]
RMDQ Roland-Morris Disability Questionnaire	Bodes et al. (2018) [36], Gül et al. (2021) [37], Orhan et al. (2021) [41]; Rabiei et al. (2021) [43], Ryan et al. (2010) [44], Téllez-Garcia et al. (2015) [47]
RSES—Rosenberg Self-Esteem Scale	Serrat et al. (2020) [46]
SF-36—Short Form Health Survey	Serrat et al. (2020) [46], Van Oosterwjck et al. (2013) [49]
SWLS—Satisfaction With Life Scale	Kohns et al. (2020) [38]
TSK/TSK-CFS—Tampa Scale of Kinesiophobia	Bodes et al. (2018) [36], Gül et al. (2021) [37], Kohns et al. (2020) [38], Malfliet et al. (2018) [39], Meeus et al. (2010) [40], Orhan et al. (2021) [41], Pires et al. (2015) [42], Ryan et al. (2010) [44], Saracoglu et al. (2020) [45], Serrat et al. (2020) [46], Téllez-Garcia et al. (2015) [47], Van Oosterwjck et al. (2013) [49]
VAS—Visual Analogue Scale	Gül et al. (2021) [37], Pires et al. (2015) [42], Rabiei et al. (2021) [43], Serrat et al. (2020) [46]

**Table 4 ijerph-20-04098-t004:** Risk of bias.

	Random Sequence Generation (Selection Bias)	Allocation Concealment (Selection Bias)	Blinding of Participants and Personnel (Performance Bias)	Blinding of Outcome Data (Attrition Bias)	Incomplete Outcome Data (Attrition Bias)	Selective Reporting (Reporting Bias)	Other Bias
Barrenengoa-Cuadra et al. [35], in 2021	LOW RISK	LOW RISK	HIGH RISK	HIGH RISK	LOW RISK	HIGH RISK	LOW RISK
Bodes et al. [36], in 2018	LOW RISK	LOW RISK	HIGH RISK	HIGH RISK	LOW RISK	HIGH RISK	LOW RISK
Gül et al. [37], in 2021	LOW RISK	UNCLEAR RISK	HIGH RISK	HIGH RISK	UNCLEAR RISK	UNCLEAR RISK	UNCLEAR RISK
Kohns et al. [38], in 2020	LOW RISK	LOW RISK	HIGH RISK	UNCLEAR RISK	LOW RISK	LOW RISK	UNCLEAR RISK
Malfliet et al. [39], in 2018	LOW RISK	LOW RISK	HIGH RISK	LOW RISK	UNCLEAR RISK	LOW RISK	UNCLEAR RISK
Meeus et al. [40], in 2010	LOW RISK	LOW RISK	HIGH RISK	LOW RISK	LOW RISK	LOW RISK	UNCLEAR RISK
Orhan et al. [41], in 2021	LOW RISK	UNCLEAR RISK	HIGH RISK	UNCLEAR RISK	LOW RISK	LOW RISK	UNCLEAR RISK
Pires et al. [42], in 2015	LOW RISK	LOW RISK	HIGH RISK	LOW RISK	LOW RISK	LOW RISK	UNCLEAR RISK
Rabiei et al. [43], in 2021	LOW RISK	LOW RISK	HIGH RISK	LOW RISK	LOW RISK	LOW RISK	UNCLEAR RISK
Ryan et al. [44], in 2010	LOW RISK	LOW RISK	HIGH RISK	HIGH RISK	LOW RISK	LOW RISK	HIGH RISK
Saracoglu et al. [45], in 2020	LOW RISK	LOW RISK	HIGH RISK	LOW RISK	LOW RISK	LOW RISK	UNCLEAR RISK
Serrat et al. [46], in 2020	LOW RISK	LOW RISK	HIGH RISK	UNCLEAR RISK	UNCLEAR RISK	LOW RISK	HIGH RISK
Téllez-Garcia et al. [47], in 2015	LOW RISK	LOW RISK	HIGH RISK	LOW RISK	LOW RISK	LOW RISK	HIGH RISK
Van Ittersum et al. [48], in 2014	LOW RISK	LOW RISK	HIGH RISK	LOW RISK	LOW RISK	LOW RISK	UNCLEAR RISK
Van Oosterwjck et al. [49], in 2013	LOW RISK	LOW RISK	HIGH RISK	HIGH RISK	LOW RISK	LOW RISK	UNCLEAR RISK

**Table 5 ijerph-20-04098-t005:** Agreement for full-text selection.

Agreement for Full-Text Selection		Evaluator 1 (B.L.)		Total
		Positive Evaluation	Negative Evaluation	
Evaluator 2 (V.B.)	Positive evaluation	10	3	13
	Negative evaluation	4	47	51
Total		14	50	64

**Table 6 ijerph-20-04098-t006:** Effectiveness of PNE based on between-group comparisons.

RCT	Experimental Intervention	←	No Difference between Groups	Control Intervention	Outcome
Barrenengoa-Cuadra et al. [35], in 2021 FU: 1 month, 6 months, 12 months	PNE + Usual treatment	**FIQ** All follow-ups (*p* < 0.001) **PCS** **BPI-**Severity **BPI-**Interference **HAQ** **HADS-A** **HADS-D** All follow-ups (*p* < 0.05)		Usual treatment	FIQ BPI-SF HAQ HADS PCS-13
Bodes et al. [36], in 2018 FU: 1 month 3 months	PNE + Therapeutic exercise	**NPRS** 3 months (*p* < 0.001) **TSK-11** 1 month (*p* < 0.05) 3 months (*p* < 0.001) **PCS-13** 1 month (*p* < 0.05) 3 months (*p* < 0.001) **RMDQ** 3 months (*p* < 0.001)		Therapeutic exercise	NPRS RMDQ PCS-13 TSK-11
Gül et al. [37], in 2021 FU: 3 weeks	PNE + Physiotherapy		**VAS** 3 weeks (*p* > 0.05) **TSK-17** 3 weeks (*p* = 0.410) **RMDQ** 3 weeks (*p* > 0.05)	Physiotherapy	VAS TSK-17 RMDQ
Kohns et al. [38], in 2020 FU: 1 month 10 months	PNE	**BPI (severità)** 1 month (*p* = 0.024) **BPI (interferenza)** 1 month (*p* = 0.031) **PSCOQ** 1 month (*p* < 0.05)	**BPI** (interference and severity) 10 months (*p* > 0.05) **PROMIS** Both follow-ups (*p* > 0.05) **PSCOQ** 10 months (*p* > 0.05) **PCS-13** Both follow-ups (*p* > 0.05) **TSK-11** Both follow-ups (*p* > 0.05) **SWLS** Both follow-up (*p* > 0.05)	Health Behavior Control	BPI PROMIS PSOCQ PCS-13 TSK-11 SWLS
Malfliet et al. [39], in 2018 FU: 2 weeks	PNE	**PDI** 2 weeks (*p* < 0.001) **PCS** 2 weeks (rumination, helplessness, magnification) (*p* < 0.001) **TSK-17** 2 weeks (*p* < 0.001) **IPQr** 2 weeks (from *p* < 0.001 to *p* = −0.01)	**PVAQ** 2 weeks (*p* > 0.05)	Neck/Back school education	PDI PCS TSK-17 IPQr PVAQ
Meeus et al. [40], in 2010 FU: Post-session	PNE	**PCS** (ruminating) Post-session (*p* = 0.009) **PCI** (distraction) Post-sessione (*p* = 0.021) **PCI** (worrying) Post-session (*p* = 0.011)	**PCS** (magnification and helplessness), Post-session (*p* > 0.01) **PCI** (transforming, reducing demands, retreating, resting) Post-session (*p* > 0.01) **TSK-CFS** Post-session (*p* > 0.01)	Pacing and self-management education	PCI PCS-13 TSK-CFS
Orhan et al. [41], in 2021 FU: 1 weeks 4 weeks	PNE “culture-sensitive”		Both follow-ups: **NRS** (*p* > 0.05) **RMDQ** (*p* > 0.05) **PBQ** (*p* > 0.05) **PCS-13** (*p* > 0.05) **TSK-17** (*p* > 0.05)	PNE standard	NRS RMDQ PBQ PCS TSK-17
Pires et al. [42], in 2015 FU: 6 weeks 3 months	PNE + Aquatic exercises	**VAS** 3 months (*p* < 0.05)	**TSK-13** (*p* > 0.05) Both follow-ups **QBPDS-PT** 6 weeks (*p* = 0.83) and 3 months (*p* = 0.09) **VAS** 6 weeks (*p* = 0.14)	Aquatic exercises	VAS QBPDS-PT TSK-13
Rabiei et al. [43], in 2021 FU: 8 weeks	PNE + Motor control exercise	**VAS** 8 weeks (*p* = 0.041) **RMDQ** 8 weeks (*p* = 0.021)	**FABQ** 8 weeks (*p* > 0.05) **PSEQ** 8 weeks (*p* > 0.05)	Group-based exercise	VAS RMDQ FABQ-W FABQ-PA PSEQ
Ryan et al. [44], in 2010 FU: 8 weeks 3 months	PNE	**NRS** 8 weeks (*p* = 0.025) **PSEQ** 8 weeks (*p* = 0.024)	**TSK-13** Both follow-ups (*p* > 0.05) **RMDQ** Both follow-ups (*p* > 0.05) **NRS** 3 months (*p* > 0.05) **PSEQ** 3 months (*p* > 0.05)	PNE + Aerobic exercise	NRS RMDQ TSK-13 PSEQ-10
Saracoglu et al. [45], in 2020 FU: 4 weeks 12 weeks	Group 1: PNE + Manual therapy + Home exercise	**NPRS** Both follow-ups: Group 1 vs. G. 3 (*p* < 0.001) **ODI** Both follow-ups: Group 1 vs. G. 3 (*p* < 0.001) **TSK-17** Both follow-ups: Group 1 vs. G. 2 (*p* < 0.001) Group 1 vs. G.3 (*p* < 0.001)	**NPRS** Both follow-ups: Group 1 vs. G. 2 (*p* = 0.01) **ODI** Both follow-ups: Group 1 vs. G.2 (*p* = 0.67)	Gruppo 2: Manual therapy + Home exercise Gruppo 3: Home exercise	NPRS ODI TSK-17
Serrat et al. [46], in 2020 FU: 6 weeks 12 weeks	Treatment as usual (TAU) + NAT-FM (PNE, Therapeutic exercise, activities in nature, Cognitive Behavioral Therapy (CBT), Mindfulness training)	**FIQR** Both follow-ups: (*p* < 0.001) **VAS** pain 6 weeks (*p* = 0.003), 12 weeks (*p* < 0.001) **VAS** fatigue 6 weeks (*p* = 0.002), 12 weeks (*p* < 0.001) **HADS-A** Both follow-ups: (*p* < 0.001) **HADS-D** 6 weeks (*p* = 0.027), 12 weeks (*p* < 0.001) **SF-36** 6 weeks (*p* = 0.017) and 12 weeks (*p* < 0.001) **TSK-11** Both follow-ups: (*p* < 0.001) **PCS** Both follow-ups: (*p* > 0.001)	**RSES** Both follow-ups: (*p* > 0.05)	Treatment as usual (TAU)	FIQR VAS (pain) VAS (fatigue) HADS-A HADS-D SF-36 TSK-17 PCS RSES
Téllez-Garcia et al. [47], in 2015 FU: 4 weeks	PNE + Dry Needling	**TSK-17** 4 weeks (*p* = 0.008) MCID > 8	**RMDQ** 4 weeks (*p* = 0.111) **ODI** 4 weeks (*p* = 0.542) **NPRS** 4 weeks (*p* = 0.801)	Dry Needling	NPRS ODI RMDQ TSK-17
Van Ittersum et al. [48], in 2014 FU: 6 weeks 6 months	PNE		**PCS** Both follow-ups: (*p* > 0.05) **FIQ** (*p* > 0.05) for both follow-ups **IPQ-R FM** (*p* > 0.05) for both follow-ups	Relaxation Education (RE)	FIQ IPQ-FM PCS
Van Oosterwjck et al. [49], in 2013 FU: 2 weeks 3 months	PNE	**SF-36** “physical functioning” (*p* = 0.046) “general health perceptions” (*p* < 0.001) “vitality” (*p* = 0.047) “mental health” (*p* < 0.001)	**FIQ** 2 weeks (*p* > 0.05) 3 months (*p* = 0.079) **PCS** Both follow-ups (rumination) (*p* = 0.219), (magnification) (*p* = 0.109), (helplessness) (*p* = 0.265) and total (*p* = 0.158) **TSK** Both follow-ups (*p* = 0.360) **PVAQ** 2 weeks (*p* > 0.05) 3 months (*p* = 0.279) **PCI** (all subdomains) Both follow-ups (*p* > 0.05) **SF-36** (other subdomains) (*p* > 0.05)	Education about pacing self-management techniques.	FIQ PCI PCS PVAQ TSK-17 SF-36

## Data Availability

Not applicable.

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
