# Peer review of "Effectiveness of Pain Neuroscience Education in Patients with Chronic Musculoskeletal Pain and Central Sensitization: A Systematic Review"

_ijerph, 2023, doi:10.3390/ijerph20054098_

Round 1

Reviewer 1 Report

Congratulations to the authors for addressing, in a systematic review, a topical issue in the interdisciplinary approach to chronic pain that has been little researched to date. However, the work has some important methodological and content limitations. Some comments that may help to remedy some of these aspects are listed below.

As a general recommendation, it would be beneficial for the authors to use the PRISMA 2020 checklist to detect and correct those aspects of the review that are not compliant.

PAGE 1

Line 13

"To investigate" does not seem a very appropriate verb for the objective of a systematic review. It would be more appropriate to "collect the available evidence", "collect and synthesize the research..."

Line 16

"RCTs" As this is the first time it appears in the text, it should be preceded by the full words, followed by the acronym, in parentheses.

Line 34

When there are more than two consecutive numbers of bibliographic citations, the proper form of citation is [2-4].

Lines 40-44

It would be helpful to provide more citations to support this statement.

PAGE 2

Lines 45-48

It is not possible to make this claim by providing only one bibliographic citation from 2011.

Line 61

In the "Background" section, central sensitization should be conceptualized because it is a prominent component of this review. Chronic musculoskeletal pain should also be conceptualized in more depth, what duration must it have to be considered chronic, what other criteria does it have to meet?

Line 72

Revise the verb of the objective, along the same lines as already proposed for the abstract.

PAGE 3

Line 94

What type of assessment was performed?

Line 101

It cannot be an exclusion criterion that the subjects are under 18 years old if, already in the inclusion criteria, it has been considered that they have to be over 18 years old.

Line 104

Any type of surgical procedure, including minor surgery and/or surgery of organs, apparatus or systems other than the musculoskeletal system?

Line 114

Were any eligibility criteria based on date of publication considered?

Line 118

PEDro and CINAHL are very discipline-specific databases (Physiotherapy and Nursing, respectively). For future reviews, take into consideration other databases such as Web of Science (WOS), Scopus, etc.

Line 127

The PRISMA 2020 Declaration states that the complete search strategies of all databases should be published, not only in one of them (as in the previous regulation). This aspect should be modified in Appendix A.

PAGE 4

Line 141

The PRISMA 2020 Statement recommends also including an assessment of the level of evidence. Please include this subsection in both the Methodology and Results.

Line 157

If the PRISMA 2020 Statement is followed, the flowchart in the PRISMA 2020 Statement should be used, which is not the one used for Figure 1.

The exclusion criteria that have been applied and the number of articles excluded for each of them should be specified.

Line 161

It says that the articles in the sample correspond to citations 22-36 and, in the data extraction table, there are two articles with citation numbers 37 and 38.

What was the final review sample, 15 or 17 articles?

Line 165

Table 1.

In the first column does not seem to fit the study design, it is rather a column for the identification of the work (author, year, country). The study design should be in a separate column.

The data collected in Table 1 should be more synthesized.

Do not include bibliographic citations in the text of the table (Ministry of Health, 2011),....

The format of the table needs to be improved.

PAGE 10

Line 170

This subsection could be expanded by specifying the smallest sample size and the study/s that had the largest sample size.

PAGE 11

Line 185

It is logical that all participants were over 18 years of age because that is an inclusion criterion for the review.

What was the mean age and age range of the patients in the overall sample? What was the distribution, by gender, of the cumulative sample?

PAGE 11

Line 198

What does "sensitized" mean in reference to an NSP intervention?

Line 204

Although it is specified in the table, it would be convenient to synthesize here the most Con CSI high profiles.

Line 226

Table 3 could be enriched with a column including the studies that used each of the measures.

The names of the columns should be added to this table.

PAGE 12

Line 227

This subsection should include a table with the results, for each of the studies included in the review, of the Risk of Bias (RoB) assessment tool of the Cochrane Collaboration, which they have said, in the Methodology section, that they had used to assess the risk of bias.

AGE 13

Line 249

Table 5 should specify the values of the pre- and post-intervention measures and not only the value of statistical significance.

What does it mean, in table 5, "ad entrambi"?

What does it mean, in table 5, "con CSI alto"?

PAGE 16

Line 265

"Bodes et al.[23], in 2018" or "Bodes et al.[23]), in the year 2018,..." This is the way to cite when naming the author. Revise throughout the paper to remove the lead author's initials in citations.

PAGE 17

Line 296

In the "Discussion" section, citations should be included to discuss the results with other authors. In this case only the articles that make up the review sample are cited again.

Only in line 392 are 4 new articles (39 to 42) cited and all of them to support the same idea.

Line 297

The objective of a systematic review is not to investigate, it is to collect and synthesize evidence from other investigations.

Author Response

Dear Editor and Dear Reviewers,

I have made all revisions as requested.

Best regards.

Valerio barbari

Reviewer 2 Report

General comments

The purpose of this paper is to measure the effectiveness of Pain Neuroscience Education (PNE) based on pain, disability, and psychosocial factors in patients with Chronic Musculoskeletal (CMS) pain and Central Sensitization (CS)

This paper could give insight that PNE application in various modalities applied, revealed that one-to-one oral sessions have a high effectiveness compared to group-or online - or written-based

This paper is related to the previous authors’ research that focuses on the effectiveness of PNE application and MSK pain

This paper has been written well, but the format for the text alignment, especially for tables and figures, needs to be revised. And several words need to recheck

Specific comments

Line 16 page: “and only RCTs enrolling”

# Consider adding “.” Before “and”

# The abbreviation of RCT hasn’t been described

Line 35-36: “the estimated prevalence is still high 35 ranging from 11.4% and 24% [5]”

# The reference has been more than 10 years. is there any reason why to use this as a reference for data estimation

Line 38-39: “indeed, it is one of the most underdiagnosed and underestimated diseases.”

# The previous paragraph states that MSK is one of the main problems in the world. But why on this sentence states that this disease is the most underdiagnosed, the most underestimated

Line 52: “written format such as booklets or telephone calls and email [10,11]”

# The references from more than 10 years ago. Is there any reason why using these?

Line 62-70: “Up to date, PNE seems to be an effective intervention for patients with persistent MSK pain, especially if associated with manual therapy and exercise [14], but nothing is known about the effectiveness of PNE in specific populations of patients with persistent MSK pain due to CS. In such scenario, most of inclusion criteria of previous primary or secondary studies were limited to the general chronic MSK pain, but no specific elegibility criteria for CS have been addressed. Since such criteria have never been specifically adopted and since chronic MSK pain does not necessarily mean CS [13], the effectiveness of PNE in patients with chronic MSK pain due to a dominance of CS pain mechanism still remains a grey area of scientific literature.”

# This paragraph is long but only has two references

Line 66: ” ….but no specific elegibility”

# It should be “….but not specific eligibility”

Line 80-81: “Only randomized controlled trials (RCTs) published in Italian or in English were considered elegible”

# Is there any reason or reference that only RCT method that eligible?

# It should be ".... considered eligible"

Line 80-81: “Quantitative Sensory Testing (QST) positive scores for CS or any other criteria”

# What is the positive score range or parameter of QST to pass the test?

Line 97: ”CSI it-self or….”

# It should be “CSI itself or….”

Line 127: ”The full search strategy for PubMed is available in the Appendix A.”

# Consider putting it into the same paragraph as the previous one.

Line 132-133: ”…the main two authors (B.L. and V.B.). and a third author (L.S.) not involed in screening process was involved in case of disagreements”

# It should be “…the main two authors (B.L. and V.B.). And a third author (L.S.) not involved in the screening process was involved in case of disagreements.”

Line 157: ”The full selection process is reported in Figure 1.”

# Consider putting it into the same paragraph as the previous one.

Line 159: Figure 1

# Consider adding a description of what the flow chart refers to.

Line 165: Table 1

# The text should align left or center to be consistent with other table or figure description

Line 176: Table 2, column 3 (Control Group), row 11 “0 0”

# What does “0 0” value mean?

Line 205: “3.1.8 Type of control groups”

# The text should align left.

Line 226: Table 3

# There is no outcome data column in the table.

Line 243: Table 4

# The text should align left or center to be consistent with other table or figure description.

Line 230: “…high risk in all studies included”

# How many total studies were included?

Line 243: Table 5

# The text is not aligned with others.

Line 290-293: “participants were divided into three groups. There was a significant improvement only in TSK-17 scores at both follow-ups in the first group (PNE, manual therapy, exercises at home) against the second group (manual therapy and exercises at home). In the first group compared with the third one (home exercises) ….”

# How many participants for each group? And based on table 5, the period of RCT is different. How does the researcher analyze the result

Line 299-302: ”Risk of bias in most of studies was low, except for the performance bias criterion rated as high risk in all studies. Nevertheless, since a low risk for performance bias may be hard to obtain in physical therapy trials, it does not seem to be a significant factor for downgrade the overall quality of evidence.”

# Consider putting it into the same paragraph as the previous one or add tab space on the first word.

Line 303: ”Overall, PNE…”

# Add tab space before the sentence to match with the other format.

Line 328-333: “For chronic spinal pain patients [26], results with a low risk of bias supporting the effectiveness of PNE over neck/back school education are limited at the unique short-term follow-up (2 weeks) and no long-term benefits were assessed. Although CSI scores were 330 not assessed at the end the study, it is notherworthy that there has been an improvement in outcomes closely related to CS, such as kinesiphobia and perception of disease, regardless of the level of CSI used to divide the participants in the baseline.”

# This paragraph is long but only has one reference

# The word “follow-up”, ’’noteworthy”, and “kinesiphobia” should be “follow-ups”, “noteworthy”, and “kinesiophobia”

Line 349-352: “Such findings are not suprising, since it is unlikely that a single PNE session lasting for 20-25 minutes with a 3-minute instructional video (experimental intervention) to be more effective that the same procedure (control intervention) without significant differences in terms of educational contents”

# The word “suprising”, and “be more effective that” should be “surprising”, and “be more effective than.”

Line 381: ”more effective throughtout”

# “throughtout” should be “throughout”

Line 392-393: ”… to recognize all those phychosocial factors…”

# “phychosocial” should be “psychosocial”

Line 445: ” In patients with FM….”

# Add tab space before the sentence to match with the other format

Reviewer 3 Report

Line 37-38; please provide the references.

The introduction part should give the background of PNE and CS. How to identify / measurement/ definition for example. 

Line 83; please provide the definition of MSK and exclusion criteria of MSK.

Line 157; Regarding the flow chart, please provide the reasons for excluded.

Line 174; Regarding the drop-out and lots to follow-up, why did the authors report it and how importance are there. 

overall the definition of MSK pain, PNE, CS are unclear. In addition, the diagnosis from persistent MSK pain are unclear.

Round 2

Reviewer 3 Report

The revised manuscript is accepted.